# Consistent text-to-image generation via Scene De-Contextualization

**Song Tang[1,2,3], Peihao Gong[1], Kunyu Li[4], Kai Guo[1], Boyu Wang[5,6], Mao Ye[7,*],**
**Jianwei Zhang[2] & Xiatian Zhu[8,*]**

[1]University of Shanghai for Science and Technology, [2]Universität Hamburg, [3]ComOriginMat Inc, [4]Fudan University,
[5]Western University, [6]Vector Institute, [7]University of Electronic Science and Technology of China, [8]University of Surrey

Project URL: https://github.com/tntek/SDeC

## Abstract

Consistent text-to-image (T2I) generation seeks to produce identity-preserving images of the same subject across diverse scenes, yet it often fails due to a phenomenon called identity (ID) shift. Previous methods have tackled this issue, but typically rely on the unrealistic assumption of knowing all target scenes in advance. This paper reveals that a key source of ID shift is the native correlation between subject and scene context, called *scene contextualization*, which arises naturally as T2I models fit the training distribution of vast natural images. We formally prove the near-universality of this scene-ID correlation and derive theoretical bounds on its strength. On this basis, we propose a novel, efficient, training-free prompt embedding editing approach, called **Scene De-Contextualization (SDeC)**, that imposes an inversion process of T2I's built-in scene contextualization. Specifically, it identifies and suppresses the latent scene-ID correlation within the ID prompt's embedding by quantifying SVD directional stability to adaptively re-weight the corresponding eigenvalues. Critically, SDeC allows for per-scene use (one scene per prompt) without requiring prior access to all target scenes. This makes it a highly flexible and general solution well-suited to real-world applications where such prior knowledge is often unavailable or varies over time. Experiments show that SDeC markedly enhances identity preservation while maintaining scene diversity.

## 1 Introduction

Text-to-image (T2I) generation (Shi et al., 2024; Saharia et al., 2022; Ramesh et al., 2021) aims to synthesize visually compelling and semantically faithful images from prompts. From artistic design to personalized media production, T2I models such as GAN (Tao et al., 2022) and Stable Diffusion (Rombach et al., 2022b) have demonstrated remarkable capability in producing novel scenes that align closely with user intent. However, in narrative-driven visual tasks involving recurring characters or entities, such as animation/video (Lei et al., 2025), personalized storytelling (Avrahami et al., 2024), cinematic pre-visualization (Tao et al., 2024), and digital avatars (Wang et al., 2023), mere alignment with scene descriptions is insufficient: The subject's IDentity (ID) must remain consistent across generated images (no ID shift). Against this backdrop, consistent T2I generation has recently emerged as a focal point of growing interest (Höllein et al., 2024; Wang et al., 2024).

Methodologically, existing approaches reduce ID shift in line with the paradigm of transfer learning (Tang et al., 2024): Extracting invariance from given heterogeneous data. This requires prior knowledge of the complete target scenes, enabling the generative model to map different scene prompts into corresponding features (Zhou et al., 2024; Liu et al., 2025) or image pseudo labels (Avrahami et al., 2024; Akdemir & Yanardag, 2024) for constructing such a diversified dataset. In practice, however, target scenes are not always available[1], rendering this assumption unrealistic and limiting

---

*Corresponding authors: Mao Ye and Xiatian Zhu

[1]In real-world projects (e.g., films, games, or story creation), the full set of final scenes, their content, and their order are often refined and finalized over numerous iterative changes, making it impossible to know all

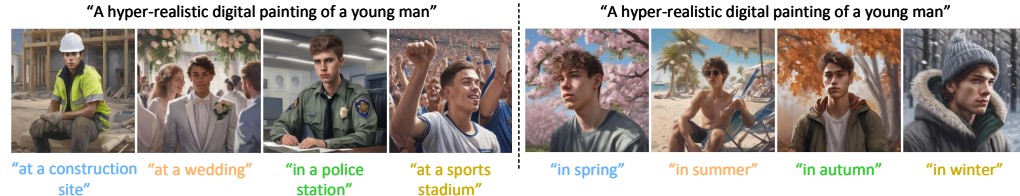

Figure 1: Illustration of scene contextualization with SDXL. **Left**: The attire of the subject varies with the site. **Right**: The subject's clothing changes with the season.

the practical applicability of these methods. Critically, the underlying cause of ID shift with T2I models remains largely unclear.

In this work, we consider that the scene plays a context role that would influence the characterization of identity, called *scene contextualization* (Fig. 1), causing ID shift. This arises because a T2I model is predominantly trained on natural images with a specific data distribution (e.g., cows often appear on the green fields but not in the sea). Consequently, the generated images are constrained to satisfy such internalized priors.

To probe the connection between scene contextualization and ID shift, we formulate a theoretical framework, showing that the contextualization, inherently induced by the attention mechanism, is not only the primary source of ID shift but also inevitable for pre-trained T2I models. Moreover, we derive theoretical bounds on contextualization strength. Building on these insights, we propose a **Scene De-Contextualization** (**SDeC**) approach, which effectively realizes the inverse process of scene contextualization. Specifically, SDeC quantifies the directional stability of the subspace spanned by Singular Value Decomposition (SVD) (Stewart, 1993) eigenvectors via a forward-and-backward eigenvalue optimization. After that, the latent scene-ID correlation within the ID prompt's embedding is identified through eigenvalue variations and then suppressed with adaptive eigenvalue weighting. The contextualization-mitigated ID prompt embedding is then reconstructed from the reweighted eigenvalues for subsequent generation. It can work in a one-prompt-per-scene setting, removing reliance on full target scenes.

Our **contributions** are: (1) We propose a *scene contextualization* perspective for ID shift with T2I models; (2) We theoretically characterize and quantify this contextualization, leading to a novel SDeC approach for mitigating ID shift per scene without the need for complete target scenes in advance. (3) Extensive experiments show that SDeC can enhance identity preservation, maintain scene diversity, and offer plug-and-play flexibility at per-scene level and across diverse tasks, e.g., integrating pose map and personalized photo, and generative backbones such as PlayGround-v2.5, RealVisXL-V4.0, Juggernaut-X-V10, SD3, and Flux.

## 2 RELATED WORK

The study of consistent T2I generation falls into two phases. The early phase focuses on *personalized T2I generation* (Zhang et al., 2024), where one or a couple of reference images are given to define the identity of interest. These methods tackle ID shift by injecting ID semantics of reference images into a pre-trained T2I model, essentially adopting two strategies. The first leverages cross-attention to progressively inject the information, subject to convergence toward the reference image(s) (Ye et al., 2023). The second is a textual token creation strategy: Transforming the reference image(s) into dedicated tokens to differentiate ID and scene. For example, DreamBooth (Ruiz et al., 2023) and variants (Sun et al., 2025; Hsin-Ying et al., 2025) introduce a unique identifier token to represent the ID and inject it by fine-tuning the generative model. PhotoMaker (Li et al., 2024) fuses the reference image(s) and text embeddings to enhance ID tokens. Textual Inversion (Gal et al., 2022) and its variants (Zeng et al., 2024; Wu et al., 2024) directly create a concept token via textual inversion.

---

subsequent scene contexts in advance. Efficiency dictates generating images online based on the current scene description, without the need for repeatedly re-generating all previous scenes (avoiding exponential complexity).

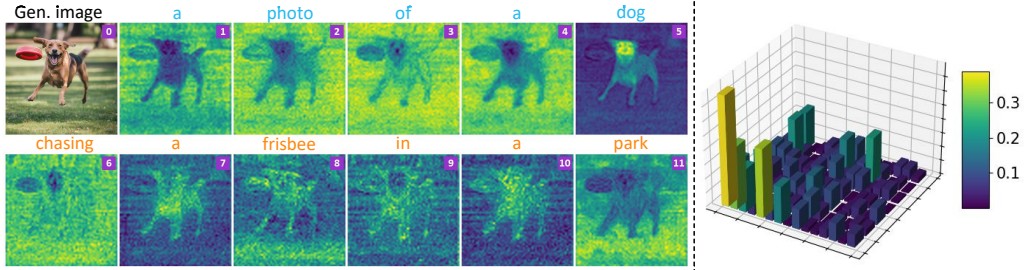

Figure 2: Example of scene contextualization with ID prompt "a photo of a dog" and scene prompt "chasing a frisbee in a park". **Left**: Correlation between tokens and attention similarity matrix shows that scene tokens affect ID generation (visualization follows (Hertz et al., 2023)). **Right**: Similarity between SVD eigenvectors of ID and scene embedding shows they share an overlapping subspace.

The recent phase moves to the more flexible *reference image-free* setting addressed under two approaches. The first involves *pseudo-label-based self-learning*: Generating candidate images with the entire prompt set, then filtering them with obvious ID shift using some metric (e.g., mutual information in ORACLE (Akdemir & Yanardag, 2024) or clustering in Chosen-One (Avrahami et al., 2024) or self-diffusion in DiffDis (Cai et al., 2025)), further retraining the generative models. Clearly, such methods are expensive due to model retraining. More recent attempts thus emerge in a *training-free* fashion. Storytelling methods (Rahman et al., 2023; Zhou et al., 2024; Tewel et al., 2024) leverage the self-attention mechanism over generated images as an adapter, whilst the state of the art, 1Prompt1Story (Liu et al., 2025), introduces a prompt re-structuring idea to highlight ID and balance scene's contribution in prompting, with extra need to couple a specific adapter. Commonly, all the methods above assumes the availability of all the scenes which often is not valid in real applications. Importantly, these studies fail to provide an insight on the underlying cause of ID shift with off-the-shelf T2I models. We address these gaps by suggesting a scene contextualization perspective along with theoretical formulation and a flexible *prompt embedding editing* solution.

## 3 SCENE CONTEXTUALIZATION

**Problem** Given an ID text prompt $\mathcal{P}_{id}$ and $K$ distinct scene text prompts $\{\mathcal{P}_{sc}^k\}_{k=1}^K$, we form a set of scenario prompts $\mathcal{P} = \{\mathcal{P}^k\}_{k=1}^K$ with $\mathcal{P}^k = \mathcal{P}_{id} \oplus \mathcal{P}_{sc}^k$. Feeding each prompt into a T2I model $\mathcal{G}$ produces $K$ generated images $\{I^k\}_{k=1}^K$, where $I^k = \mathcal{G}(\mathcal{P}^k)$. Consistent T2I generation requires that $\{I^k\}_{k=1}^K$ simultaneously (1) preserve the same identity features specified by $\mathcal{P}_{id}$, and (2) faithfully reflect the scene semantics described in each corresponding $\mathcal{P}^k$.

**Interaction between ID and scene** The attention mechanism is the key structure in Transformer based T2I models. Denote $Z \in \mathbb{R}^{N \times d}$ as the set of $N$ input token features of dimension $d$, the self-attention projections are: $K = ZW_K$, $V = ZW_V$, $Q = ZW_Q$. For any query $q \in Q$, the attention output is computed as:

$$O(q) = \alpha^\top V \text{ with } \alpha = \text{softmax}\left( (qK^\top)/\sqrt{d_k} \right). \tag{1}$$

where $d_k$ denotes the feature dimension with $W_K$. In the prompt embedding space, this attention operation offers an opportunity for scene tokens to inject its context information into ID tokens, potentially leading to ID shift. We name this *scene contextualization*.

For intuitive understanding, we visualize the correlation between scene tokens and ID tokens by their cross-attention similarity matrix, where each row corresponds to $\alpha$ values. As shown in images # 7∼10 of Fig. 2-Left, the bright regions (green/yellow) within the subject (dog) clearly indicate that scene tokens affect the generation of this dog.

### 3.1 THEOREM BEHIND SCENE CONTEXTUALIZATION

Attention operations are typically executed in a chain-like manner for T2I models. For simplicity, we analyze the first attention block in generative models, without loss of generality.

**Theorem 1** *Let $\mathcal{H}_{\mathrm{id}}, \mathcal{H}_{\mathrm{sc}} \subset \mathbb{R}^d$ be the identity/scene semantic subspaces with ideal semantic separation: $\mathcal{H}_{\mathrm{id}} \cap \mathcal{H}_{\mathrm{sc}} = \varnothing$; $\mathcal{Z}_{\mathrm{id}} \subseteq \mathcal{H}_{\mathrm{id}}$ and $\mathcal{Z}_{\mathrm{sc}}^k \subseteq \mathcal{H}_{\mathrm{sc}}$ be the prompt-embedding matrix of $\mathcal{P}_{\mathrm{id}}$ and $\mathcal{P}_{\mathrm{sc}}^k$; the prompt-embedding matrix be partitioned by semantics $\mathcal{Z} = \left[\mathcal{Z}_{\mathrm{id}}; \mathcal{Z}_{\mathrm{sc}}^k\right] \in \mathbb{R}^{n \times d}$. For any query $q_{\mathrm{id}}$ associated with the identity, its attention output can be specified to*

$$O(q_{\mathrm{id}}) = \alpha^\top (\mathcal{Z} W_V) = \alpha_{\mathrm{id}}^\top (\mathcal{Z}_{\mathrm{id}} W_V) \; + \; \alpha_{\mathrm{sc}}^\top (\mathcal{Z}_{\mathrm{sc}}^k W_V), \quad (2)$$

*where the attention weights by token index $\alpha = [\alpha_{\mathrm{id}}, \alpha_{\mathrm{sc}}]$ conforming to the row split of $\mathcal{Z}$. Let $\Pi_{\mathrm{id}}$ be the orthogonal projector onto $\mathcal{H}_{\mathrm{id}}$. The projection of $O(q_{\mathrm{id}})$ onto the identity subspace $\mathcal{H}_{\mathrm{id}}$ is*

$$\Pi_{\mathrm{id}}\left[O(q_{\mathrm{id}})\right] = \underbrace{\Pi_{\mathrm{id}}\left[\alpha_{\mathrm{id}}^\top (\mathcal{Z}_{\mathrm{id}} W_V)\right]}_{\text{id term: } T_{\mathrm{id}}} + \underbrace{\Pi_{\mathrm{id}}\left[\alpha_{\mathrm{sc}}^\top (\mathcal{Z}_{\mathrm{sc}}^k W_V)\right]}_{\text{scene term: } T_{\mathrm{sc}}}. \quad (3)$$

*Assume $\Pi_{\mathrm{id}} \circ W_V\big|_{\mathcal{H}_{\mathrm{sc}}}$ denotes an operation where an input from subspace $\mathcal{H}_{\mathrm{sc}}$ is first transformed by $W_V$ and then projected onto subspace $\mathcal{H}_{\mathrm{id}}$. If the conditions, (A) $\alpha_{\mathrm{sc}} \neq \mathbf{0}$ and (B) $\Pi_{\mathrm{id}} \circ W_V\big|_{\mathcal{H}_{\mathrm{sc}}} \neq 0$, hold, then the scene term in Eq. (3) is nonzero: $T_{\mathrm{sc}} \neq 0$.*

Theorem 1 suggests that even if $\mathcal{H}_{\mathrm{id}} \cap \mathcal{H}_{\mathrm{sc}} = \varnothing$, the attention could still cause scene contextualization. The two conditions are almost always satisfied with T2I models. Condition (A) is often met for two factors. (i) Keys from scene tokens and queries from ID tokens are rarely strictly orthogonal or sufficiently separated; so the softmax attention weights are unlikely to be exactly zero, leaving scene tokens with non-negligible attention mass. (ii) No enforcement on separating between scene and identity tokens during training, lead scene-to-ID attention positive. For condition (B), it is equivalent to *non-block-diagonality* of $W_V$ w.r.t. the decomposition $\mathcal{H}_{\mathrm{id}} \oplus \mathcal{H}_{\mathrm{scene}}$: No scene vector is mapped with a nonzero component in $\mathcal{H}_{\mathrm{id}}$ — again this condition is not enforced in training.

Theorem 1 assumes an idealized condition of $\mathcal{H}_{\mathrm{id}} \cap \mathcal{H}_{\mathrm{sc}} = \varnothing$. In practice, ID and scene subspaces often exhibit partial overlap. To assess this, we apply SVD to $\mathcal{Z}_{\mathrm{id}}$ and $\mathcal{Z}_{\mathrm{sc}}^k$ and compute the similarity between their corresponding eigenvectors. As seen in Fig. 2-Right, the high regions in the similarity matrix reveal nontrivial correlations between $\mathcal{Z}_{\mathrm{id}}$ and $\mathcal{Z}_{\mathrm{sc}}^k$ across certain dimensions. Relaxing the disjoint-subspace assumption to this correlated case, we show that scene contextualization still persists below.

**Corollary 1** *Assume that $\mathcal{H}_{\mathrm{id}}$ and $\mathcal{H}_{\mathrm{sc}}$ have a nontrivial intersection: $\mathcal{H}_\cap := \mathcal{H}_{\mathrm{id}} \cap \mathcal{H}_{\mathrm{sc}}$ with $k_\cap := \dim(\mathcal{H}_\cap) > 0$ where $\dim(\cdot)$ means space dimensions. If $\alpha_{\mathrm{sc}} \neq 0$, then for a generic linear mapping $W_V$, which excludes measure-zero degenerate cases, $T_{\mathrm{sc}} \neq 0$ hold.*

The degenerate case above refers to the rare weight setting where $W_V$ maps the scene subspace $\mathcal{H}_{\mathrm{sc}}$ exactly onto a subspace orthogonal to $\mathcal{H}_{\mathrm{id}}$. Such cases form a measure-zero set in the parameter space. With random initialization and continuous optimization, the probability of encountering them is negligible. For a typical $W_V$, this blocking effect does not arise. Moreover, unlike the idealized assumptions in Theorem 1, in practice $k_\cap > 0$, meaning scene tokens always receive some attention. This makes contextualization both easier and stronger, even when $W_V$ only weakly couples $\mathcal{H}_{\mathrm{sc}}$ and $\mathcal{H}_{\mathrm{id}}$. Please see the proof in `Appendix-B`.

Combining Theorem 1 and Corollary 1 yields an insight that, irrespective of whether $\mathcal{H}_{\mathrm{id}}$ and $\mathcal{H}_{\mathrm{sc}}$ overlap, the scene-to-ID projection is generically nonzero, i.e., scene contextualization occurs firmly.

### 3.2 BOUNDING THE STRENGTH OF CONTEXTUALIZATION

In this section, we derive a bound on contextualization strength, uncovering the key variables that govern its intensity. Theoretically, we characterize the scene contextualization to $T_{\mathrm{sc}}$ (see Theorem 1), thereby its spectral norm $\big\|T_{\mathrm{sc}}\big\|_2$ can be a measurement of strength. $\big\|T_{\mathrm{sc}}\big\|_2$ can be bounded as below (refer to the proof in `Appendix-C`):

**Theorem 2** *Let $P_\cap$ be the orthogonal projector onto $\mathcal{H}_\cap$; $\Pi_{\mathrm{sc}}$ be the orthogonal projector onto $\mathcal{H}_{\mathrm{sc}}$; $P_\cap^\perp := \Pi_{\mathrm{sc}} - P_\cap$ be the projector onto the orthogonal complement within $\mathcal{H}_{\mathrm{sc}}$. Define $R_\cap := \mathcal{Z}_{\mathrm{sc}}^k P_\cap$, $R_\perp := \mathcal{Z}_{\mathrm{sc}}^k P_\cap^\perp$, $T_\cap := \Pi_{\mathrm{id}} W_V P_\cap$, $T_\perp := P_\cap^\perp W_V \Pi_{\mathrm{id}}$, and $\epsilon := \|\alpha_{\mathrm{sc}}^\top\|_2$. The contextualization strength $\big\|T_{\mathrm{sc}}\big\|_2$ is bounded as*

$$0 \leq \|T_{\mathrm{sc}}\|_2 \; \leq \; \epsilon \cdot \|R_\cap\|_2 \cdot \|T_\cap\|_F \; + \; \epsilon \cdot \|R_\perp\|_2 \cdot \|T_\perp\|_F. \quad (4)$$

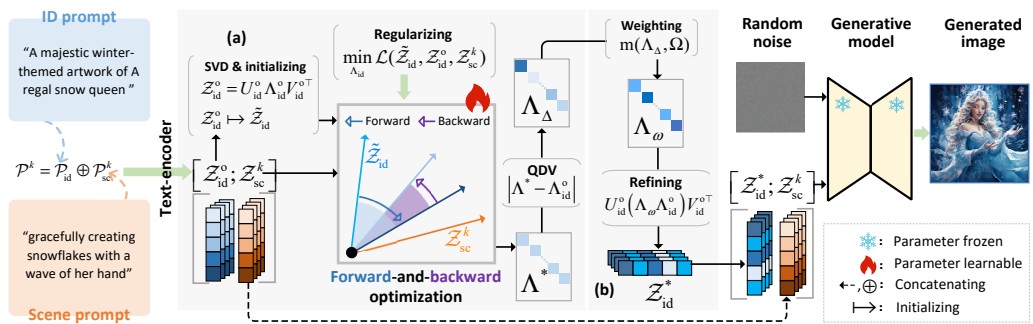

Figure 3: Overview of SDeC. A text prompt $\mathcal{P}^k = \mathcal{P}_{\mathrm{id}} \oplus \mathcal{P}_{\mathrm{sc}}^k$ is encoded into prompt embeddings $[\mathcal{Z}_{\mathrm{id}}^{\mathrm{o}}; \mathcal{Z}_{\mathrm{sc}}^k]$ where $^{\mathrm{o}}$ means "original". SDeC mitigates scene contextualization by (a) identifying and (b) suppressing latent scene-ID correlation in $\mathcal{Z}_{\mathrm{id}}^{\mathrm{o}}$ (QDV: Quantifying Directional influence Variations). The refined ID embedding $\mathcal{Z}_{\mathrm{id}}^*$ is then concatenated with $\mathcal{Z}_{\mathrm{sc}}^k$ for subsequent generation.

In Theorem 2, $\Pi_{\mathrm{id}}$ is formally a subspace operator. In practice, it is often spanned or approximately defined by the ID embedding $\mathcal{Z}_{\mathrm{id}}$ itself. Editing $\mathcal{Z}_{\mathrm{id}}$ is thus equivalent to adjusting the orientation of the subspace, thereby modifying $\Pi_{\mathrm{id}}$. Following this, we derive a corollary to further specify the upper bound from the ID perspective (see the proof in `Appendix-D`).

**Corollary 2** $\mathcal{H}_{\mathrm{id}}$ *is the subspace spanned by the ID embedding* $\mathcal{Z}_{\mathrm{id}}$; $U$ *is an orthonormal basis of* $\mathcal{H}_{\mathrm{id}}$, *i.e.,* $U = \mathrm{orth}(\mathcal{Z}_{\mathrm{id}})$, *where* $\mathrm{orth}(\cdot)$ *specifies an orthogonalization operation. Writing the projector onto the ID subspace as* $\Pi_{\mathrm{id}} = UU^\top$, *we have*

$$0 \le \|T_{\mathrm{sc}}\|_2 \le \epsilon \cdot \|R_\cap\|_2 \cdot \underbrace{\|U^\top W_V P_\cap\|_F}_{\sigma_\cap} + \epsilon \cdot \|R_\perp\|_2 \cdot \underbrace{\|W_V^\top P_\cap^\perp U\|_F}_{\sigma_\perp}. \tag{5}$$

In Corollary 2, $\sigma_\cap$ measures the energy shared between ID ($U$) and scene ($\mathcal{Z}_{\mathrm{sc}}^k$) subspaces, while $\sigma_\perp$ denotes the energy of ID projected into the scene-specific subspace. We consider that the majority of scene-ID interaction takes place via $\sigma_\cap$, whilst $\sigma_\perp$ allows for holistic coherence. Thus, we only need to minimize $\sigma_\cap$. This decomposition further reveals a balance between lowering scene–ID interaction and maintaining coherence, indicating an inherent disentanglement–coherence trade-off.

## 4 Scene De-Contextualization

**Overview** Grounding on the insights from Corollary 2, we propose the SDeC framework to achieve de-contextualization, including (1) estimating $P_\cap$ and (2) driving $\sigma_\cap \to 0$. SDeC's idea is to quantify the extent to which each direction is influenced by contextualization and then selectively reinforce those that are less affected. Thus, the original ID embedding is edited. Here, the directions that are strongly affected are referred to as the *latent scene–ID correlation subspace*.

Concretely, we approximate $P_\cap$ by identifying the latent scene-ID correlation subspace in $\mathcal{Z}_{\mathrm{id}}$ (Fig. 3(a)), exploiting a learning process. We then suppress this subspace (Fig. 3(b)) to reach $\sigma_\cap \to 0$. For clarity, we hereafter denote $\mathcal{Z}_{\mathrm{id}}$ to $\mathcal{Z}_{\mathrm{id}}^{\mathrm{o}}$ and in-training ID embedding is denoted as $\tilde{\mathcal{Z}}_{\mathrm{id}}$.

**Identifying latent scene-ID correlation subspace in $\mathcal{Z}_{\mathrm{id}}^{\mathrm{o}}$** We first achieve the directional correlation measurement via an "forward-and-backward" optimization (Fig. 3 (a)): First pulling $\tilde{\mathcal{Z}}_{\mathrm{id}}$ closer to the scene $\mathcal{Z}_{\mathrm{sc}}^k$ (forward), followed by restoring back to its original position $\mathcal{Z}_{\mathrm{id}}^{\mathrm{o}}$ (backward). We start with solving $\mathcal{Z}_{\mathrm{id}}^{\mathrm{o}} = U_{\mathrm{id}}^{\mathrm{o}} \Lambda_{\mathrm{id}}^{\mathrm{o}} V_{\mathrm{id}}^{\mathrm{o}\top}$ by SVD. Let $\tilde{\mathcal{Z}}_{\mathrm{id}} = U_{\mathrm{id}}^{\mathrm{o}} \Lambda_{\mathrm{id}} V_{\mathrm{id}}^{\mathrm{o}\top}$ with $\Lambda_{\mathrm{id}}$ initiated as $\Lambda_{\mathrm{id}}^{\mathrm{o}}$; We formulate a two-phase optimization problem:

$$\Lambda^* = \min_{\Lambda_{\mathrm{id}}} \mathcal{L}(\tilde{\mathcal{Z}}_{\mathrm{id}}, \mathcal{Z}_{\mathrm{id}}^{\mathrm{o}}, \mathcal{Z}_{\mathrm{sc}}^k) = \| U_{\mathrm{id}}^{\mathrm{o}} \Lambda_{\mathrm{id}} V_{\mathrm{id}}^{\mathrm{o}\top} - \mathcal{Z}_{\mathrm{sc}}^k \|_2 + \beta \| U_{\mathrm{id}}^{\mathrm{o}} \Lambda_{\mathrm{id}} V_{\mathrm{id}}^{\mathrm{o}\top} - \mathcal{Z}_{\mathrm{id}}^{\mathrm{o}} \|_2,$$

$$\beta = 0 \text{ when } iter \le M, \text{ and } \beta \ne 0 \text{ when } iter > M, \tag{6}$$

where $iter$ denotes the training iteration index; $M$ denotes the length of the first training phase, correspondingly, iterations $0 \sim M$ correspond to the forward process, while the remaining iterations constitute the backward process.

In the two-phase optimization defined in Eq. (6), the forward phase identifies the directions in $\mathcal{Z}_{\mathrm{id}}$ that align with $\mathcal{Z}_{\mathrm{sc}}^k$, capturing their shared representations. However, some of these directions may also be essential for representing $\mathcal{Z}_{\mathrm{id}}$ itself. To mitigate potential semantic degradation in $\mathcal{Z}_{\mathrm{id}}$, we introduce a backward phase that progressively recovers these ID-associated components.

After evaluating the directional correlations, we further quantify their strength. In the SVD view, the directions, whose eigenvalues remain nearly unchanged (resistant to both pull and restoration), correspond to robust directions against contextualization. In contrast, those with large variations are the latent scene-ID correlation subspace, which theoretically corresponds to the $P_{\cap}$. In this context, we use *absolute spectral excursion* to quantify the stability in different directions. Assume $\Lambda^*$ and $\Lambda_{\mathrm{id}}^{\mathrm{o}}$ share the same spectral structure (diagonal with ordered singular values): $\Lambda^{(\cdot)} = \mathrm{diag}(\lambda_1^{(\cdot)}, \dots, \lambda_r^{(\cdot)})$ with $\lambda_1^{(\cdot)} \geq \cdots \geq \lambda_r^{(\cdot)} \geq 0$ and $r = \mathrm{rank}(\mathcal{Z}_{\mathrm{id}}^{\mathrm{o}})$. The directional correlation of $\mathcal{Z}_{\mathrm{id}}^{\mathrm{o}}$ can be formulated by Eq. (7) where $v_i$ stands for the correlation strength of the $i$-th direction.

$$\Lambda_\Delta = |\Lambda^* - \Lambda_{\mathrm{id}}^{\mathrm{o}}| = \mathrm{diag}(v_1, \dots, v_i, \dots, v_r) \text{ with } v_i = |\lambda_i^* - \lambda_i^{\mathrm{o}}|. \tag{7}$$

**De-contextualization by suppressing latent scene-ID correlation subspace in $\mathcal{Z}_{\mathrm{id}}^{\mathrm{o}}$**  We achieve this through robust subspace filtering, which involves eigenvalue modulation, denoted by $\mathrm{m}(\cdot, \cdot)$, followed by reconstructing the ID prompt embedding using the modulated eigenvalues. This method features (1) relative enhancement on the robust subspace, and (2) soft direction selection without threshold. Let $\mathcal{Z}_{\mathrm{id}}^*$ be the refined ID prompt embedding. The filtering is expressed as

$$\mathcal{Z}_{\mathrm{id}}^* = U_{\mathrm{id}}^{\mathrm{o}} \left(\Lambda_\omega \Lambda_{\mathrm{id}}^{\mathrm{o}}\right) V_{\mathrm{id}}^{\mathrm{o}\top} \text{ with } \Lambda_\omega = \mathrm{m}(\Lambda_\Delta, \Omega) = 1 + \Omega \left( \frac{\Lambda_\Delta - \Delta_{\min}}{\Delta_{\max} - \Delta_{\min}} \right), \tag{8}$$

where $\Delta_{\max}$ and $\Delta_{\min}$ are the maximum and minimum entries of $\Lambda_\Delta$, respectively, and the hyperparameter $\Omega \geq 1$ controls the weighting strength. In Eq. (8), the weighting values $\Lambda_\omega \in [1, 1+\Omega]$ are derived from normalized directional influence. Setting $\Omega \geq 1$ ensures that the robust subspace is emphasized while avoiding semantic loss in the shared subspace.

After filtering, we edit the original prompt embedding $[\mathcal{Z}_{\mathrm{id}}^{\mathrm{o}}; \mathcal{Z}_{\mathrm{sc}}^k]$ to $\mathcal{Z}^{k*} = [\mathcal{Z}_{\mathrm{id}}^*; \mathcal{Z}_{\mathrm{sc}}^k]$. We feed $\mathcal{Z}^{k*}$ into the T2I model to produce the final image.

## 5  EXPERIMENTS

**Benchmark**  Our evaluation uses the ConsiStory+ (Liu et al., 2025), extending the ConsiStory dataset (Tewel et al., 2024) to 192 prompt sets, generating 1292 images with a wider range of subjects, descriptions, and styles. The scene includes **all contextual factors**: not only environmental attributes (e.g., lighting, style, background elements) but also actions, behaviors, or temporary states associated with the subject in that specific image.

**Evaluation metrics**  To assess ID consistency, we employ two metrics: (1) CLIP-I (Hessel et al., 2021), computed as the cosine distance between image embeddings, and (2) DreamSim-F (Fu et al., 2023), better aligned with human judgment of visual similarity closely. As DreamSim, we remove the background using CarveKit (Selin, 2023) and fill random noise, ensuring that similarity measurement focus solely on ID content.

To evaluate the entire scenario (ID + scene), we adopt CLIP-T, the average CLIPScore (Hessel et al., 2021) between each generated image and its corresponding prompt. Note, this metric cannot measure the undesired scene mixture effect (see Fig. 7-Middle in `Appendix-F`). To address this, we introduce a new metric, `DreamSim-B`, specifically designed to quantify inter-scene interference, in the spirit of DreamSim-F, based on foreground masking instead.

**Competitors**  We consider two types of competitors. The first is baseline T2I models, including SD1.5 (Rombach et al., 2022a) and SDXL (Podell et al., 2023). The second includes six state-of-the-art consistent T2I methods: BLIP-Diffusion (Li et al., 2023), Textual Inversion (Gal et al., 2022), PhotoMaker (Li et al., 2024), ConsiStory (Tewel et al., 2024), StoryDiffusion (Zhou et al., 2024),

Table 1: Quantitative comparison. The best and second-best results are marked in **bold** and underlined, respectively. PE: Prompt embedding Editing; POT: Prompt Operation Time (per image); GenT: Generation Time (per image).

| | Method | Base model | PE | ID metrics | | Scenario metrics | | Infer. time↓ (s) | | VRAM↓ (GB) | Steps |
|---|---|---|---|---|---|---|---|---|---|---|---|
| | | | | DreamSim-F↓ | CLIP-I↑ | DreamSim-B↑ | CLIP-T↑ | POT | GenT | | |
| Baseline | – | SD1.5 | ✗ | 0.4118 | 0.8071 | **0.4673** | 0.8324 | 0 | 2 | 3.18 | 50 |
| | – | SDXL | ✗ | 0.2778 | 0.8558 | 0.3861 | 0.8865 | 0 | 9 | 10.72 | 50 |
| Training | BLIP-Diffusion | SD1.5 | ✗ | 0.2851 | 0.8522 | 0.3957 | 0.8187 | 0 | 1 | 3.54 | 26 |
| | Textual Inversion | SDXL | ✗ | 0.3066 | 0.8437 | 0.3919 | 0.8557 | 0 | 10 | 14.00 | 40 |
| | PhotoMaker | SDXL | ✗ | 0.2808 | 0.8545 | 0.3957 | 0.8812 | 0 | 9 | 9.74 | 50 |
| Training-Free | ConsiStory | SDXL | ✗ | 0.2729 | 0.8604 | 0.4207 | 0.8942 | 0 | 27 | 15.58 | 50 |
| | StoryDiffusion | SDXL | ✗ | 0.3197 | 0.8502 | 0.4214 | 0.8578 | 0 | 24 | 38.44 | 50 |
| | 1P1S | SDXL | ✗ | **0.2238** | **0.8798** | 0.2955 | 0.8883 | 0.10 | 22 | 13.10 | 50 |
| | 1P1S w/o IPCA | SDXL | ✗ | 0.2682 | 0.8617 | 0.3338 | 0.8637 | 0.10 | 19 | 10.74 | 50 |
| | **SDeC** | SDXL | ✓ | 0.2589 | 0.8655 | 0.3675 | 0.8946 | 0.61 | 15 | 12.14 | 50 |
| | **SDeC+ConsiStory** | SDXL | ✓ | 0.2542 | 0.8744 | 0.4155 | **0.8967** | 0.67 | 27 | 15.56 | 50 |

and 1Prompt1Story (1P1S) (Liu et al., 2025). The first three are training based, vs training free for the rest and SDeC. For more extensive test, we introduce (1) 1P1S w/o IPCA, with the attention module IPCA removed to focus on its prompt embedding editing; and (2) SDeC+ConsiStory, to test the complementary effect of our method with existing adapter based method ConsiStory. We exclude IP-Adapter (Ye et al., 2023) from comparison, as its generated characters are homogeneous in pose and layout, with little ability to follow the scene description instruction. The same setting is applied to all compared methods. (see `Appendix-E.2`).

## 5.1 RESULTS ANALYSIS

**Quantitative analysis** We draw these observations from Tab. 1: (1) For training-free methods, 1P1S delivers the best result in the ID metrics, whilst suffering serious inter-scene interference (worst DreamSim-B score, also see Fig. 7 in `Appendix-F` for the typical qualitative evidence), largely unacceptable for consistent T2I generation. In contrast, our SDeC strikes the best balance between ID consistency and scene diversity. (2) Without the attention IPCA, 1P1S is outperformed by SDeC across all metrics. That indicates that our prompt embedding editing is superior, even without the need for all target scenes in advance. (3) ConsiStory lags behind SDeC in ID metrics, but excels in scene background. (4) SDeC is well complementary with ConsiStory to further push the performance, as they address distinct aspects. (5) Interestingly, training-based methods are even outpaced by most tree-free counterparts in ID consistency, with extra disadvantage in efficiency. (6) In terms of memory and inference time, SDeC introduces negligible overhead on top (see `Appendix H.1` for further discussion).

**User study** To complement those evaluation metrics, we further conduct a user study. We compare with top alternatives: PhotoMaker, ConsiStory, StoryDiffusion, and 1P1S. Specifically, the test images were generated by 30 random prompt sets from ConsiStory+. A total of 20 volunteers were invited to pick which image best balances among ID consistency,

Table 2: User study results. Criteria: Best balance in ID consistency, scene diversity, and prompt alignment.

| Method | PhotoMaker | ConsiStory | StoryDiffusion | 1P1S | **SDeC** |
|---|---|---|---|---|---|
| Wins↑ | 8.17% | 20.83% | 13.33% | 15.00% | 42.67% |

scene diversity, and prompt alignment. We measure the performance using the percentage of wins. Tab. 2 shows that SDeC best matches human preference.

**Qualitative analysis** For visual comparison, we show a couple of examples in Fig. 4. For the robotic elephant case, ConsiStory presents varying robotic styles, whilst 1P1S suffers from scene interference. For the cup of hot chocolate case, the ID shift issue becomes more acute with prior methods. Instead, SDeC still does a favored job. More qualitative results are provided in `Appendix-G`.

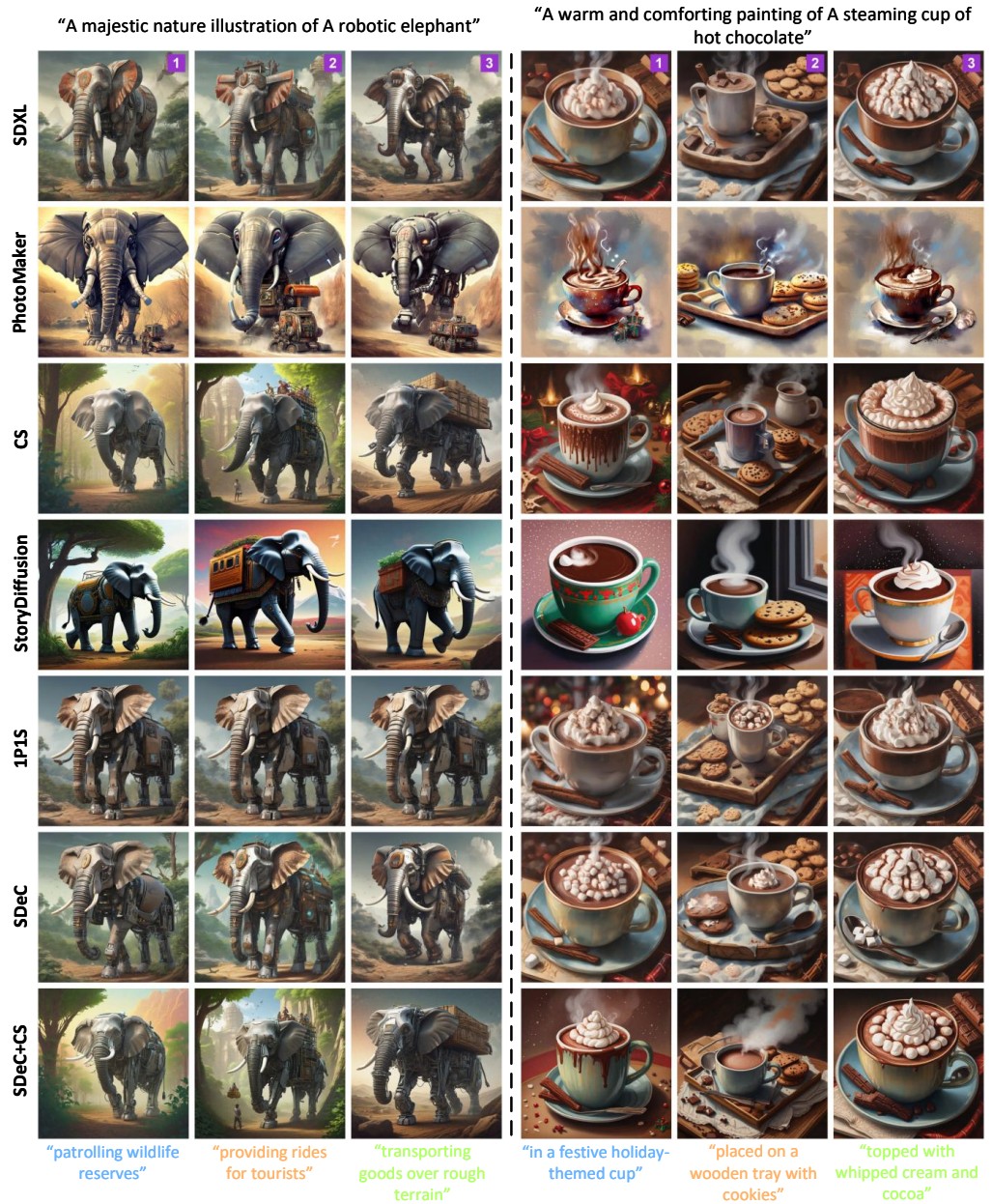

Figure 4: Qualitative results. CS: ConsiStory.

## 5.2 FURTHER ANALYSIS

**Ablation study** In SDeC, there are two key designs: (1) Estimate $P_\cap$ in a soft manner by a learning process, and (2) employ the absolute excursion of SVD eigenvalues to identify the latent scene-ID correlation subspace. To isolate their effect, we tailor two variations of SDeC: (1) SDeC w/o soft-estimation, where we operate the shared subspace in a hard way: By constructing a correlation matrix to estimate $P_\cap$ (its implementation details are provided in Appendix-E.3), and (2) SDeC w/o abs-excursion where the corresponding eigenvalue normalizes the eigenvalue variation.

Tab. 3 presents the ablation study results. SDeC ranks first in terms of ID metrics and scenario alignment (see CLIP-T), indicating that the two designs positively affect the final performance. These results are understandable. In reality, the relationship between ID and scene subspaces is complicated.

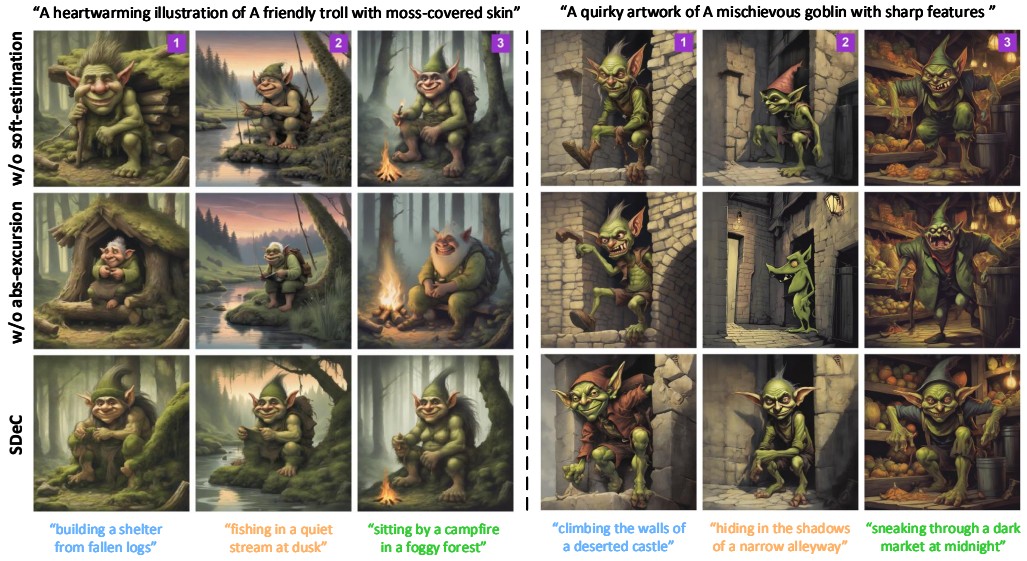

Figure 5: Qualitative comparison results of SDeC and its two variations SDeC w/o soft-estimation, SDeC w/o abs-excursion. The results of SDeC significantly outperform the variant methods. Moreover, SDeC w/o abs-excursion disregards the original importance of the SVD eigen-directions, causing greater semantics loss than both SDeC w/o soft-estimation and SDeC, and consequently ranks last in generation quality.

Thus, explicitly constructing $P_\cap$ is nearly infeasible. A tractable approach is by approximation using high-dimensional matrix decomposition, which, however, is numerically unstable under limited samples (Yang et al., 2023). Accordingly, SDeC w/o soft-estimation suffers from a performance decrease. As for SDeC w/o abs-excursion, its relative excursion strategy enforces the stability quantifying on a unified scale, disregarding the intrinsic importance of each eigen-direction.

Additionally, relative to SDeC, the $P_\cap$ estimated by these two variant approaches is inherently less reliable. When applied to contextualization carrier suppression, it introduces additional and unpredictable interference into the ID embedding. Through the coupling effect of attention, this interference propagates into the scene generation process, ultimately amplifying discrepancies across scenes. This is why the variant approaches have a higher DreamSim-B score than SDeC.

Table 3: Ablation study. Best results are in **bold**.

| Method | ID metrics | | Scene metrics | |
|---|---|---|---|---|
| | DreamSim-F↓ | CLIP-I↑ | DreamSim-B↑ | CLIP-T↑ |
| SDeC w/o soft-estimation | 0.3351 | 0.8320 | 0.4254 | 0.8755 |
| SDeC w/o abs-excursion | 0.3576 | 0.8190 | **0.4440** | 0.8569 |
| SDeC | **0.2589** | **0.8655** | 0.3675 | **0.8946** |

As a supplement to Tab. 3, in Fig. 5, we present a qualitative comparison between methods SDeC w/o soft-estimation, SDeC w/o abs-excursion, and SDeC. It is seen that the qualitative results are consistent with the data in Tab. 3. In particular, method SDeC w/o abs-excursion performs worse than method SDeC w/o soft-estimation, which supports our previous discussion. Specifically, SDeC w/o abs-excursion disregards the inherent importance of different SVD eigen directions, resulting in greater semantic loss and severe subject deformation (for example, see image #2 in Fig. 5-Right). In contrast, SDeC w/o soft-estimation, which employs a matrix transformation approach, just suffers from less accurate in identifying latent overlapping subspaces, but remain the key semantic information. Thus, it achieves better ID preservation than SDeC w/o abs-excursion.

**Validation of scene contextualization control** Our validation is based on a scenario with ID prompt: "A mischievous fantasy depiction of A cunning goblin with sharp features" and scene prompt: "trading stolen trinkets at a market" where "trinkets" is sliced to "trin" and "kets" by text-

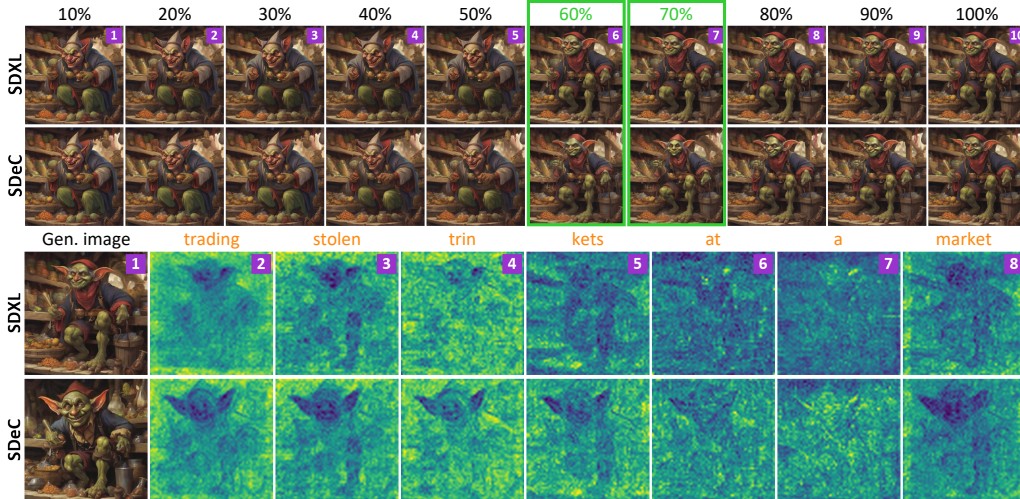

Figure 6: Validation of scene contextualization control. **Top**: PCA-based analysis where the image pairs blocked by the green-highlighted box have noticeable discrepancy. **Bottom**: Correlation between scene token embedding and attention similarity matrix.

encoder. Employing SVD to the original ID prompt embedding, we have $\mathcal{Z}_{\text{id}}^{\text{o}} = U_{\text{id}}^{\text{o}} \Lambda_{\text{id}}^{\text{o}} V_{\text{id}}^{\text{o}\top}$. For the refined one $\mathcal{Z}_{\text{id}}^{*}$ obtained by our SDeC, the weighting coefficients is $\Lambda_{\omega}$ (see Eq.(8)).

The key to SDeC is that the detected scene-ID correlation directions can modulate identity traits. To verify this, we apply PCA-based suppression to $\mathcal{Z}_{\text{id}}^{\text{o}}$ using criteria $\Lambda_{\text{id}}^{\text{o}}$ and $\Lambda_{\omega}$, respectively. Sweeping the cumulative energy threshold in 10% increments yields the ten image pairs shown in Fig. 6-Top. In pairs #6–7, the subject produced by SDeC noticeably diverges from its SDXL counterpart, indicating that SDeC indeed identifies overlapping directions that drive identity adjustment.

Additionally, our prompt editing design operates through the attention module. Accordingly, the most direct evidence of contextualization control is to examine the correlation between token embeddings and the attention similarity matrix. Fig. 6-Bottom visualizes it as $\mathcal{Z}_{\text{id}}^{*}$ drives the generation. With bright green denoting high correlation, we observe that, except for image #7, SDeC yields darker regions with more pronounced subject silhouettes than SDXL. These observations suggest that SDeC reduces the scene contextualization and provides an intuitive explanation for SDeC's effectiveness. The more model analyses are provided in `Appendix-H` and `Appendix-I`.

**Analysis of a proprietary commercial product** Google's latest flagship T2I product, Nano Banana (Google & DeepMind, 2025), has demonstrated impressive ID-preservation ability. While we cannot integrate our method, we design an interesting test to inspect how this system might work, which may reveal additional insights for future work. The results and speculative analysis are presented in `Appendix-J`.

## 6 CONCLUSION

In this paper, we identify scene contextualization as a key source of ID shift in T2I generation and conduct a formal investigation. Our analysis shows that this contextualization is an inevitable attention-induced phenomenon and the primary driver of ID shift in T2I models. By deriving theoretical bounds on its strength, we provide a foundation for mitigating this effect. Building on these insights, we introduce SDeC, a training-free embedding editing method that suppresses latent scene-ID correlation subspace through eigenvalue stability analysis, yielding refined ID embeddings for more consistent generation. Extensive experiments validate both the effectiveness and generality of the proposed approach. The limitations and future work are elaborated in `Appendix-K`.

ACKNOWLEDGMENTS

This work is supported by the National Natural Science Foundation of China (62476169), the Sichuan Science and Technology Program (2025YFMS0024), and the UKRI-AHRC CoSTAR National Lab for Creative Industries Research and Development (AH/Y001060/1).

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

## A PROOF OF THEOREM 1

**Restatement of Theorem 1** *Let $\mathcal{H}_{id}, \mathcal{H}_{sc} \subset \mathbb{R}^d$ be the identity/scene semantic subspaces, which have ideal semantic separation: $\mathcal{H}_{id} \cap \mathcal{H}_{sc} = \varnothing$. Let $\mathcal{Z}_{id} \subseteq \mathcal{H}_{id}$ and $\mathcal{Z}_{sc}^k \subseteq \mathcal{H}_{sc}$ be the prompt-embedding matrix of $\mathcal{P}_{id}$ and $\mathcal{P}_{sc}^k$, respectively; the prompt-embedding matrix be partitioned by semantics $\mathcal{Z} = \left[ \mathcal{Z}_{id}; \mathcal{Z}_{sc}^k \right] \in \mathbb{R}^{n \times d}$. For any query $q_{id}$ associated with the identity, its attention output can be specified to*

$$O(q_{\text{id}}) = \alpha^\top (\mathcal{Z} W_V) = \alpha_{\text{id}}^\top (\mathcal{Z}_{\text{id}} W_V) + \alpha_{\text{sc}}^\top (\mathcal{Z}_{\text{sc}}^k W_V), \tag{9}$$

*where the attention weights by token index $\alpha = [\alpha_{\text{id}}, \alpha_{\text{sc}}]$ conforming to the row split of Z. Let $\Pi_{\text{id}}$ be the orthogonal projector onto $\mathcal{H}_{\text{id}}$. The projection of $O(q_{\text{id}})$ onto the identity subspace $\mathcal{H}_{\text{id}}$ is*

$$\Pi_{\text{id}} \left[ O(q_{\text{id}}) \right] = \underbrace{\Pi_{\text{id}} \left[ \alpha_{\text{id}}^\top (\mathcal{Z}_{\text{id}}^k W_V) \right]}_{\text{id term: } T_{\text{id}}} + \underbrace{\Pi_{\text{id}} \left[ \alpha_{\text{sc}}^\top (\mathcal{Z}_{\text{sc}}^k W_V) \right]}_{\text{scene term: } T_{\text{sc}}}. \tag{10}$$

*Assume $\Pi_{\text{id}} \circ W_V \big|_{\mathcal{H}_{\text{sc}}}$ denotes an operation where an input from subspace $\mathcal{H}_{\text{sc}}$ is first transformed by $W_V$ and then projected onto subspace $\mathcal{H}_{\text{id}}$. If condition (A) $\alpha_{\text{sc}} \neq \mathbf{0}$ and (B) $\Pi_{\text{id}} \circ W_V \big|_{\mathcal{H}_{\text{sc}}} \neq 0$ hold, then the scene term in Eq. (10)) is nonzero. That is, $T_{\text{sc}} \neq 0$.*

**Proof 1** *To obtain the result presented above, we need to firstly prove (1) $O_{\text{sc}}(q_{\text{id}}) = \alpha_{\text{sc}}^\top \left( \mathcal{Z}_{\text{sc}}^k W_V \right) \neq 0$ (the second term in Eq. (9)), and (2) its projection onto space $\mathcal{H}_{\text{sc}}$ is non-zero.*

**(1) Proving $O_{\text{sc}}(q_{\text{id}}) \neq \mathbf{0}$.** *Due to $\alpha_{\text{sc}} \neq \mathbf{0}$, $O_{\text{sc}}$ is a non-trivial linear combination of the rows of $Z_{\text{sc}} W_V$; hence, it is nonzero unless $\mathcal{Z}_{\text{sc}} W_V$ vanishes row-wise, which is not the case in functioning models. This establishes that a nonzero scene term $O_{\text{sc}}$ is present in $\text{Att}(q_{\text{id}})$.*

**(2) Proving the projection of $O_{\text{sc}}(q_{\text{id}})$ onto $\mathcal{H}_{\text{id}}$ is non-zero.** *From The condition $\Pi_{\mathcal{H}_{\text{id}}} \circ W_V \big|_{\mathcal{H}_{\text{sc}}} = \mathbf{0}$, for any scene vector $z^{(s)} \in \mathcal{H}_{\text{sc}}$, we have $\Pi_{\mathcal{H}_{\text{id}}} (W_V z^{(s)}) \neq \mathbf{0}$. Then, the scene contribution has a nonzero component along $\mathcal{H}_{\text{id}}$. Because $\alpha$ is a softmax, all its entries are non-negative and at least one scene weight is strictly positive (since $\alpha_{\text{sc}} \neq \mathbf{0}$). Therefore,*

$$\begin{aligned} T_{\text{sc}} &= \Pi_{\text{id}} \left[ \alpha_{\text{sc}}^\top (\mathcal{Z}_{\text{sc}} W_V) \right] \\ &= \sum_{j \in \text{scene}} \alpha_j \, \Pi_{\text{id}} \left[ (z_j^{(s)})^\top W_V \right] \neq \mathbf{0}. \end{aligned} \tag{11}$$

## B PROOF OF COROLLARY 1

**Restatement of Corollary 1** *Assume $\mathcal{H}_{\text{id}}$ and $\mathcal{H}_{\text{sc}}$ have a nontrivial intersection: $\mathcal{H}_{\cap} := \mathcal{H}_{\text{id}} \cap \mathcal{H}_{\text{sc}}$, $k_{\cap} := \dim(\mathcal{H}_{\cap}) > 0$ where $\dim(\cdot)$ means space dimensions. If $\alpha_{\text{sc}} \neq 0$, then for a generic linear mapping $W_V$, which excludes measure-zero degenerate cases, $T_{\text{sc}} \neq 0$ hold.*

**Proof 2** *Pick any nonzero $u \in \mathcal{H}_{\cap}$. Since $u \in \mathcal{H}_{\text{sc}}$ and $\mathcal{Z}_{\text{sc}} \supseteq \mathcal{H}_{\text{sc}}$, there exists a scene row $z^{(s)}$ such that $z^{(s)} = \beta u + z^\perp$ with $\beta \neq 0$, $z^\perp \perp \mathcal{H}_{\text{id}}$. For any $W_V$, we further have*

$$\Pi_{\text{id}}(W_V z^{(s)}) = \beta \, \Pi_{\text{id}}(W_V u) + \Pi_{\text{id}}(W_V z^\perp). \tag{12}$$

*Consider the set $\mathcal{U} := \{ W_V : \Pi_{\text{id}}(W_V u) = 0 \}$, which is a proper linear subspace of $\mathbb{R}^{d \times d}$ and hence has Lebesgue measure zero. Therefore, for almost every $W_V$, we have $\Pi_{\text{id}}(W_V u) \neq 0$, implying $\Pi_{\text{id}}(W_V z^{(s)}) \neq 0$. Since $\alpha_{\text{sc}} \neq 0$ has at least one strictly positive entry,*

$$\begin{aligned} T_{\text{sc}} &= \Pi_{\text{id}} \left[ \alpha_{\text{sc}}^\top (\mathcal{Z}_{\text{sc}} W_V) \right] \\ &= \sum_{j \in \text{scene}} \alpha_j \, \Pi_{\text{id}} (W_V z_j^{(s)}) \neq 0. \end{aligned} \tag{13}$$

## C  PROOF OF THEOREM 2

**Restatement of Theorem 2** *Let $P_\cap$ be the orthogonal projector onto $\mathcal{H}_\cap$; $\Pi_{\mathrm{sc}}$ be the orthogonal projector onto $\mathcal{H}_{\mathrm{sc}}$; $P_\cap^\perp := \Pi_{\mathrm{sc}} - P_\cap$ be the projector onto the orthogonal complement within $\mathcal{H}_{\mathrm{sc}}$. Define $R_\cap := \mathcal{Z}_{\mathrm{sc}}^k P_\cap$, $R_\perp := \mathcal{Z}_{\mathrm{sc}}^k P_\cap^\perp$, $T_\cap := \Pi_{\mathrm{id}} W_V P_\cap$, $T_\perp := P_\cap^\perp W_V \Pi_{\mathrm{id}}$, and $\epsilon := \|\alpha_{\mathrm{sc}}^\top\|_2$. For contextualization strength $\left\|T_{\mathrm{sc}}\right\|_2$, it is bounded by*

$$0 \le \|T_{\mathrm{sc}}\|_2 \ \le \ \epsilon \cdot \|R_\cap\|_2 \cdot \|T_\cap\|_F \ + \ \epsilon \cdot \|R_\perp\|_2 \cdot \|T_\perp\|_F. \tag{14}$$

**Proof 3** *Since $\mathrm{col}(\mathcal{Z}_{\mathrm{sc}}^k) \subseteq \mathcal{H}_{\mathrm{sc}}$, for any vector $x$, $\mathcal{Z}_{\mathrm{sc}}^k x = \mathcal{Z}_{\mathrm{sc}}^k \Pi_{\mathrm{sc}} x = \mathcal{Z}_{\mathrm{sc}}^k (P_\cap + P_\cap^\perp)x$. Thus*

$$\begin{aligned} T_{\mathrm{sc}} &= \Pi_{\mathrm{id}} \big[ \alpha_{\mathrm{sc}}^\top (\mathcal{Z}_{\mathrm{sc}}^k W_V) \big] \\ &= \alpha_{\mathrm{sc}}^\top (\mathcal{Z}_{\mathrm{sc}}^k W_V \Pi_{\mathrm{id}}) \\ &= \alpha_{\mathrm{sc}}^\top ((\mathcal{Z}_{\mathrm{sc}}^k (P_\cap + P_\cap^\perp)) W_V \Pi_{\mathrm{id}}) \\ &= \alpha_{\mathrm{sc}}^\top \mathcal{Z}_{\mathrm{sc}}^k P_\cap W_V \Pi_{\mathrm{id}} + \alpha_{\mathrm{sc}}^\top \mathcal{Z}_{\mathrm{sc}}^k P_\cap^\perp W_V \Pi_{\mathrm{id}}. \end{aligned} \tag{15}$$

*Based on $R_\cap := Z_{\mathrm{sc}}^k P_\cap$, $R_\perp := Z_{\mathrm{sc}}^k P_\cap^\perp$, $T_\cap := \Pi_{\mathrm{id}} W_V P_\cap$, $T_\perp := P_\cap^\perp W_V \Pi_{\mathrm{id}}$, Eq. (15) becomes*

$$\begin{aligned} T_{\mathrm{sc}} &= \alpha_{\mathrm{sc}}^\top \mathcal{Z}_{\mathrm{sc}}^k P_\cap P_\cap W_V \Pi_{\mathrm{id}} + \alpha_{\mathrm{sc}}^\top \mathcal{Z}_{\mathrm{sc}}^k P_\cap^\perp P_\cap^\perp W_V \Pi_{\mathrm{id}} \\ &= \alpha_{\mathrm{sc}}^\top R_\cap T_\cap + \alpha_{\mathrm{sc}}^\top R_\perp T_\perp. \end{aligned} \tag{16}$$

*Applying the triangle inequality ($\|a + b\| \le \|a\| + \|b\|$) to Eq. (16), we have the following inequality.*

$$\|T_{\mathrm{sc}}\|_2 \ \le \ \|\alpha_{\mathrm{sc}}^\top R_\cap T_\cap\|_2 + \|\alpha_{\mathrm{sc}}^\top R_\perp T_\perp\|_2. \tag{17}$$

*For any row vector $x^\top$ and matrix $A$, $\|x^\top A\|_2 \le \|x\|_2 \|A\|_F$ (column-wise Cauchy–Schwarz). Also $\|YZ\|_2 \le \|Y\|_2 \|Z\|_2$ (submultiplicativity). Apply these to each term in Eq. (17) to obtain*

$$\begin{aligned} \|\alpha_{\mathrm{sc}}^\top R_\cap T_\cap\|_2 &\le \|\alpha_{\mathrm{sc}}^\top\|_2 \cdot \|R_\cap\|_2 \cdot \|T_\cap\|_F = \epsilon \cdot \|R_\cap\|_2 \cdot \|T_\cap\|_F, \\ \|\alpha_{\mathrm{sc}}^\top R_\perp T_\perp\|_2 &\le \|\alpha_{\mathrm{sc}}^\top\|_2 \cdot \|R_\perp\|_2 \cdot \|T_\perp\|_F = \epsilon \cdot \|R_\perp\|_2 \cdot \|T_\perp\|_F. \end{aligned} \tag{18}$$

*Summing the two inequalities above gives the upper bound claim.*

## D  PROOF OF COROLLARY 2

**Restatement of Corollary 2** *$\mathcal{H}_{\mathrm{id}}$ is the subspace spanned by the ID embedding $\mathcal{Z}_{\mathrm{id}}$; $U$ is an orthonormal basis of $\mathcal{H}_{\mathrm{id}}$, i.e., $U = \mathrm{orth}(\mathcal{Z}_{\mathrm{id}})$, where $\mathrm{orth}(\cdot)$ means performing orthogonalization on the input. Writing the projector onto the ID subspace as $\Pi_{\mathrm{id}} = UU^\top$, we have*

$$0 \le \|T_{\mathrm{sc}}\|_2 \ \le \ \epsilon \cdot \|R_\cap\|_2 \cdot \|U^\top W_V P_\cap\|_F \ + \ \epsilon \cdot \|R_\perp\|_2 \cdot \|W_V^\top P_\cap^\perp U\|_F. \tag{19}$$

**Proof 4** *Since $\Pi_{\mathrm{id}} = UU^\top$, the term $\|T_\cap\|_F$ and $\|T_\perp\|_F$ in Theorem 2 can be*

$$\begin{aligned} \|\Pi_{\mathrm{id}} W_V P_\cap\|_F^2 &= \|UU^\top W_V P_\cap\|_F^2 \\ \|W_V^\top P_\cap^\perp \Pi_{\mathrm{id}}\|_F^2 &= \|W_V^\top P_\cap^\perp UU^\top\|_F^2. \end{aligned} \tag{20}$$

*We first prove $\|UU^\top W_V P_\cap\|_F^2 = \|U^\top W_V P_\cap\|_F^2$. Expanding the left-hand side of it, we have*

$$\begin{aligned} \|UU^\top W_V P_\cap\|_F^2 &= \mathrm{tr}\big((UU^\top W_V P_\cap)^\top (UU^\top W_V P_\cap)\big) \\ &= \mathrm{tr}\big(P_\cap^\top W_V^\top UU^\top UU^\top W_V P_\cap\big). \end{aligned} \tag{21}$$

*Since $UU^\top$ is an orthogonal projector, we have $UU^\top UU^\top = UU^\top$; then applying the cyclic property of the trace, we obtain:*

$$\begin{aligned} \|UU^\top W_V P_\cap\|_F^2 &= \mathrm{tr}\big(P_\cap^\top W_V^\top UU^\top W_V P_\cap\big) \\ &= \mathrm{tr}\big(U^\top W_V P_\cap P_\cap^\top W_V^\top U\big). \end{aligned} \tag{22}$$

Table 4: Code link of the comparison methods.

| Method | Code link |
|---|---|
| BLIP-Diffusion (Li et al., 2023) | https://github.com/salesforce/LAVIS/tree/main/projects/blip-diffusion |
| Textual Inversion (Gal et al., 2022) | https://github.com/oss-roettger/XL-Textual-Inversion |
| PhotoMaker (Li et al., 2024) | https://github.com/TencentARC/PhotoMaker |
| ConsiStory (Tewel et al., 2024) | https://github.com/NVlabs/consistory |
| StoryDiffusion (Zhou et al., 2024) | https://github.com/HVision-NKU/StoryDiffusion |
| 1Prompt1Story (Liu et al., 2025) | https://github.com/byliutao/1Prompt1Story |

*Because $P_\cap$ is itself an orthogonal projector, we have $P_\cap^2 = P_\cap$ and $P_\cap^\top = P_\cap$. Therefore,*

$$\|UU^\top W_V P_\cap\|_F^2 = \text{tr}\big((U^\top W_V P_\cap)(U^\top W_V P_\cap)^\top\big)$$
$$= \|U^\top W_V P_\cap\|_F^2. \tag{23}$$

*In the similar way, we have $\|W_V^\top P_\cap^\perp UU^\top\|_F^2 = \|W_V^\top P_\cap^\perp U\|_F$. Substituting it and Eq. (23) to the upper bound from Theorem 2, we finish the proof.*

# E    EXPERIMENTAL DETAILS

## E.1    PARAMETERS SETTING

SDeC involves three parameters: Trade-off parameter $\beta$ and switching constant $M$ in Eq. (6) and weighting strength $\Omega$ in Eq. (8). We fix $(\beta, M) = (10, 20)$ in all experiments. For UNet-based (Ronneberger et al., 2015) generative models, we set $\Omega = 1$, whereas MMDiT-based (Esser et al., 2024) models use a larger value, $\Omega = 10$. A detailed description of the parameter-setting guidelines is provided in `Parameter Analysis` section (Appendix H.2).

## E.2    MODEL SETTING

We achieve ID-preserving image generation by editing the ID prompt embeddings during the inference phase. There is no extra training or optimization imposed on the generative models. In practice, we adopt Stable Diffusion (SD) V1.5[2] as the backbone model of BLIP-Diffusion, while the pre-trained Stable Diffusion XL (SDXL)[3] is selected as the backbone model for our SDeC and the rest of the comparison approaches.

The comparison methods are implemented using the unofficial or official codes from GitHub website, whose details are listed in Tab. 4. The computation platform adopts the same configuration as SDeC: NVIDIA RTX A6000 GPU with 48GB VRAM.

In addition, among these comparisons, BLIP-Diffusion (Li et al., 2023) and PhotoMaker (Li et al., 2024) rely on an additional reference image. We produce this reference by feeding the identity prompt into their respective base models, the same as 1Prompt1Story (1P1S) (Liu et al., 2025).

## E.3    IMPLEMENTATION DETAILS OF METHOD SDeC W/O SOFT-ESTIMATION

The goal of **SDeC w/o soft-estimation** is to find the overlapping subspace directions between $\mathscr{Z}_{\text{id}}$ and $\mathscr{Z}_{\text{sc}}$, i.e., directions corresponding to a principal angle $\theta \approx 0$. We achieve this by:

We first perform SVD on both $\mathscr{Z}_{\text{id}}$ and $\mathscr{Z}_{\text{sc}}$ to obtain their orthonormal bases:

$$\mathscr{Z}_{\text{id}} = U_{\text{id}}\Lambda_{\text{id}}V_{\text{id}}^\top, \quad \mathscr{Z}_{\text{sc}} = U_{\text{sc}}\Lambda_{\text{sc}}V_{\text{sc}}^\top. \tag{24}$$

Then, the column subspaces are constructed as

$$B_{\text{id}} := V_{\text{id}}[:, 1:r_{\text{id}}], \quad B_{\text{sc}} := V_{\text{sc}}[:, 1:r_{\text{sc}}], \tag{25}$$

---

[2]https://huggingface.co/stable-diffusion-v1-5/stable-diffusion-v1-5
[3]https://huggingface.co/stabilityai/stable-diffusion-xl-base-1.0

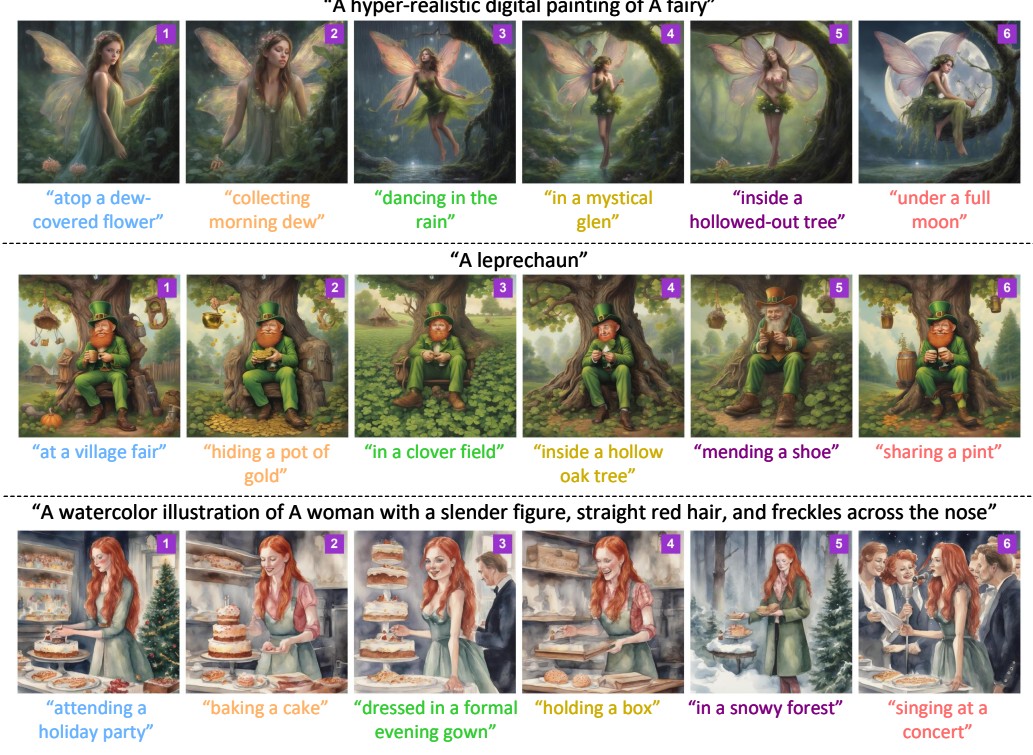

Figure 7: Demonstration of scene-level interference caused by the prompt integration strategy in 1Prompt1Story (Liu et al., 2025). The visual interference elements are: **Top**: The bent tree in images #3∼6, **Middle**: The oak tree in all images, and **Bottom**: The cake in images #1∼5.

where $r_{\mathrm{id}}$ and $r_{\mathrm{sc}}$ denote the respective subspace ranks.

Subsequently, based on $B_{\mathrm{id}}$ and $B_{\mathrm{sc}}$, we can compute their correlation matrix as

$$M = B_{\mathrm{id}}^\top B_{\mathrm{sc}} \ \in \mathbb{R}^{r_{\mathrm{id}} \times r_{\mathrm{sc}}}. \tag{26}$$

By performing SVD, we have $M = U\Lambda V^\top$, where the singular values $\sigma_i = \cos\theta_i$ correspond to the cosines of the principal angles $\theta_i$. We select $\sigma_i \geq \tau$ (with $\tau$ typically chosen 0.98), where the associated singular vectors indicate the intersection directions.

Thus, the explicit basis for the intersection subspace in the original space can be written as $B_\cap = B_{\mathrm{id}} U_{(:,\mathcal{I})}$, where $\mathcal{I} = \{i : \sigma_i \geq \tau\}$. Applying an additional QR orthonormalization on $B_\cap$ yields the final intersection basis $\hat{B}_\cap$. The projection operator onto the intersection subspace then can be expressed by $P_\cap \approx \hat{B}_\cap \hat{B}_\cap^\top$. Corresponding to the proposed scheme in SDeC, we equivalently detect the latent scene–ID correlation subspace. After that, we can realize suppressing this correlation subspace by employing $\mathcal{Z}_{\mathrm{id}}^* = \mathcal{Z}_{\mathrm{id}}(I - P_\cap)$ directly.

## F  DEMONSTRATION OF SCENE-LEVEL INTERFERENCE IN 1PROMPT1STORY

The previous 1Prompt1Story method (Liu et al., 2025) proposed a prompt strategy that merges the ID prompt with all scene prompts into a single input, followed by calibration across different scenes using SVD-based scene prompt selection. Subsequently, all images are generated from this consolidated prompt with a fixed ID component, thereby reducing ID shift. With the aid of the proposed attention module, serving as an adapter, this strategy further enhances identity preservation.

However, the proposed prompt strategy in 1Prompt1Story inevitably introduces pronounced *scene-level* interference. For an intuitive view, we demonstrate this phenomenon from 1Prompt1Story's results on the ConsiStory+ benchmark. As shown in Fig. 7, in the Top, images #1 and #2 both exhibit

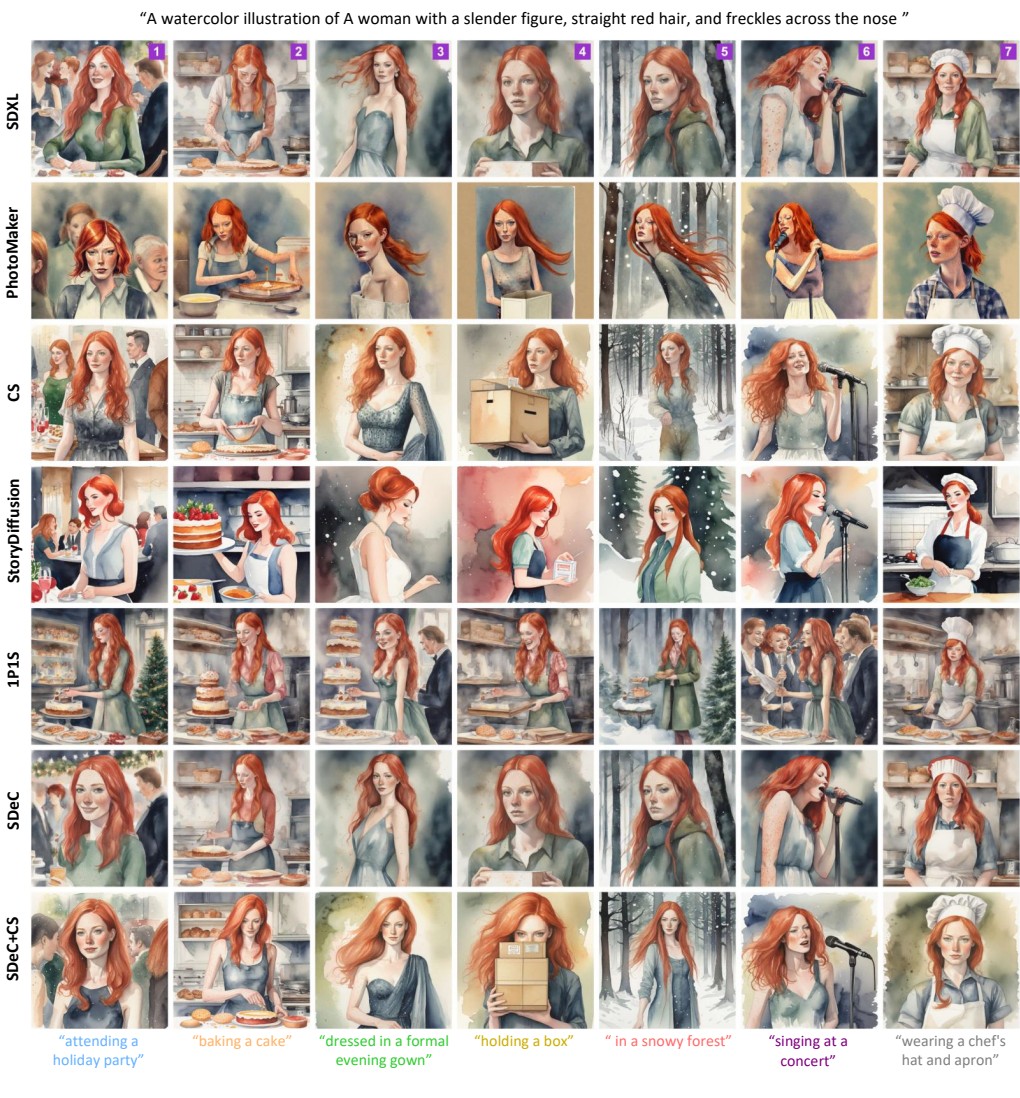

Figure 8: Case one of supplementary qualitative comparison results.

similar dense vegetation in the bottom-right corner, while the remaining images consistently feature a bent tree (mentioned in the prompt of image #5); in the Middle, all images are dominated by the visual element of an oak tree (mentioned in the prompt of image #4).

## G  MORE QUALITATIVE RESULTS

As a supplement to Fig. 4, we provide more qualitative comparison results in Fig. 8 and Fig. 9. Although without the story-wide context, the images generated by our method in a "one prompt per scene" manner present better ID preservation while well scene matching.

## H  FURTHER MODEL ANALYSIS

### H.1  DISCUSSION OF SCALABILITY

Although the SVD step in SDeC has a theoretical complexity of $O(d^3)$ ($d$ is dimension of matrix), it does not form a practical bottleneck during inference. In practice, SDeC incurs only negligible overhead, as the generative model itself accounts for more than 90% of the total computation time.

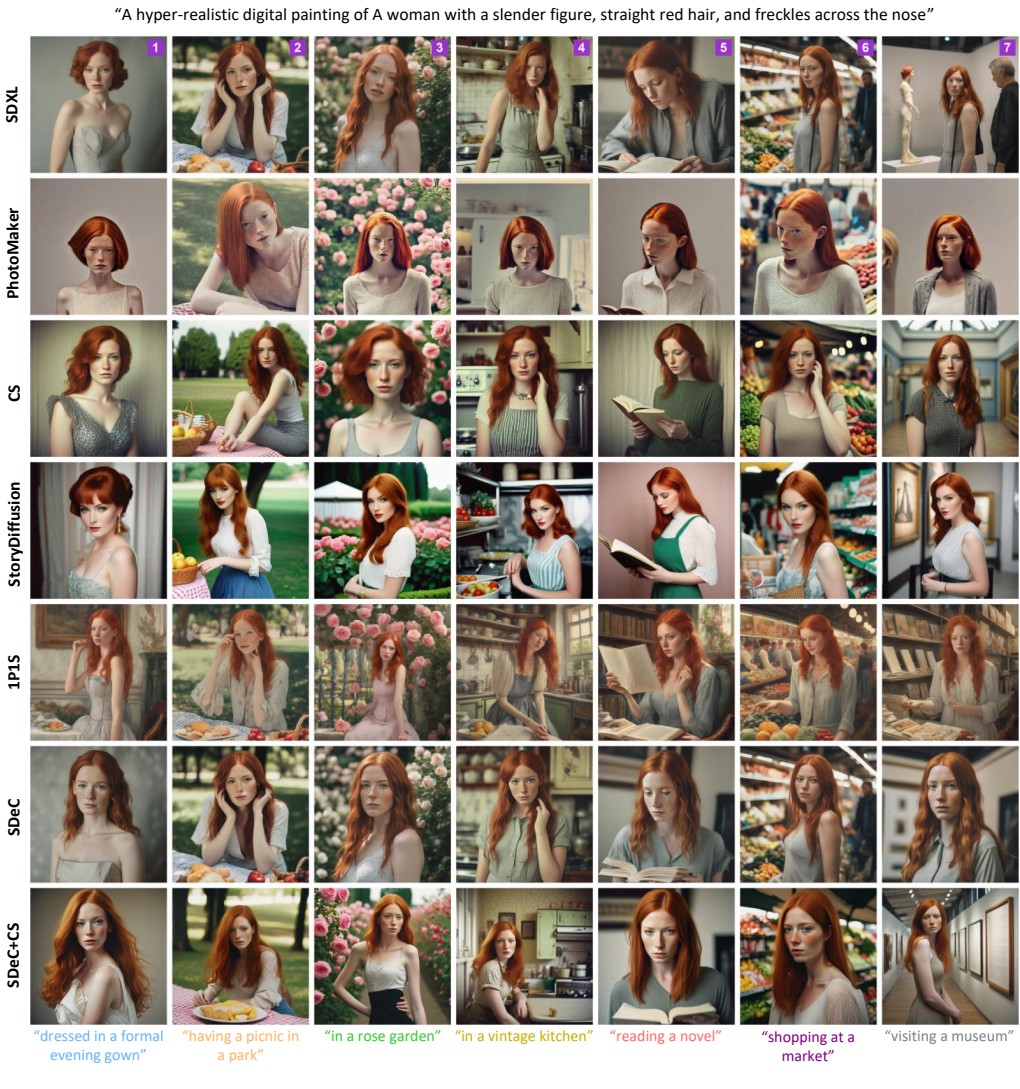

Figure 9: Case two of supplementary qualitative comparison results.

To further improve scalability, particularly for extremely long prompts, several extensions merit investigation in future work. First, *approximate SVD techniques*, such as Randomized SVD (Halko et al., 2011) with $O(d^2 \log k)$ complexity or the Nyström approximation (Williams & Seeger, 2001) with $O(dk^2)$ ($k$ is a small constant), could significantly reduce computational cost while preserving the essential spectral structure. Second, a *divide-and-conquer strategy*, in which a long prompt is partitioned into $K$ sub-prompts and processed sequentially, would allow the overhead to grow approximately linearly with $K$ (about $K \times 0.61$ seconds). These observations suggest that SDeC remains scalable and practical even for large-scale or dynamically evolving prompting scenarios.

## H.2 PARAMETER ANALYSIS

Our SDeC method involves three parameters, including weighting strength $\Omega$ in Eq. (8) and trade-off parameter $\beta$ and switching constant $M$ in Eq. (6). In this part, we first specify the setting of these hyperparameters for optimal performance, and then conduct a sensitivity analysis.

**Guidelines for parameter setting** In our SDeC method, the setting of hyperparameters for optimal performance should follow the guidelines below:

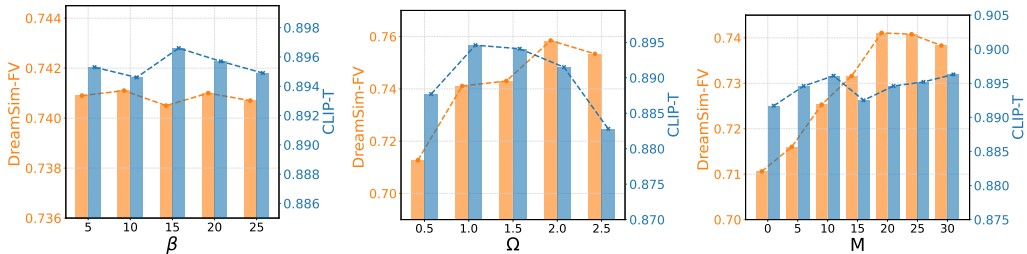

Figure 10: Parameter sensitivity analysis results of $\beta \in [5, 25]$, $\Omega \in [0.5, 2.5]$, and $M \in [0, 30]$. The DreamSim-F score is inverted to DreamSim-FV = 1-DreamSim-F, so higher values indicate better performance, as with CLIP-T score.

(1) Parameter $\beta_M$ is associated with the "forward-and-backward" optimization. Practically, $M$ should exceed a threshold (20) so that the forward phase can capture as many scene-related directions within the ID embedding as possible. Moreover, $\beta$ should satisfy $\beta \gg 0$, enabling maximal restoration of ID-related information in the backward phase. This "forward-and-backward" optimization aims to identify the latent ID-scene correlation space, independent of and generic with any specific generative model. Consequently, the setting of $\beta$ and $M$ remains consistent across different generative architectures.

(2) Parameter $\Omega$ is tied to the architecture of the underlying generative model, as it represents the strength of de-contextualization and is therefore directly related to the generation process. Our experiments indicate that for UNet-based models (e.g., SDXL, PlayGround, RealVisXL-V4.0, Juggernaut-X-V10), $\Omega$ should be small (typically $\Omega = 1$). In contrast, DiT-based models (e.g., SD3, Flux) require a much larger value of $\Omega$ (typically $\Omega = 10$).

**Sensitivity analysis** In Fig. 10, we provide the varying curves of DreamSim-F and CLIP-T scores as the three parameters change. As shown in Fig. 10-Left, the DreamSim-FV and CLIP-T scores remain near-steady when a wide range of $[5, 20]$, indicating our method does not rely on the careful selection of trade-off parameter $\beta$.

The results in Fig. 10-Middle show that $\Omega$'s increase leads to a trade-off phenomenon between the DreamSim-FV and CLIP-T score. This phenomenon is reasonable. In SDeC, to avoid semantics loss, we enhance the robust subspace by imposing larger weights $\Omega$ on its corresponding eigen-directions (see Eq. (8)). When the increase of $\Omega$ remains within a reasonable range, the native correlation between ID and scene is reduced, improving both ID preservation and scene retention. Once it exceeds a threshold (e.g., $> 1$), however, the generative model is prone to pay more attention to the stronger ID components, leading to continued gains in ID preservation but convergence in scenes.

As for parameter $M$, Fig. 10-Right presents that DreamSim-FV curve progressively climbs and reaches the top at $M = 20$, while ClIP-T curve is with only minor fluctuations. The results are consistent with our expectations. In the forward-and-backward process, sufficient time in the forth stage drives the ID prompt embedding closer to that of the scene, enabling the identification of directions in ID prompt embedding's eigen-space most sensitive to contextualization. As this process acts only on ID, scene diversity remains unaffected.

### H.3 SENSITIVITY TO THE ORDERING OF SCENE AND ID PROMPTS

In the evaluation experiments, the prompt is arranged naturally: the ID prompt precedes the scene prompt. To investigate the sensitivity to the ID-scene prompt order, we conduct an additional comparison where the scene and ID prompts are reversed. In this way, we can extend the original SDXL and SDeC (rewritten as SDXL-o and SDeC-o) to variation SDXL-r and SDeC-r, respectively.

The comparison results reported in Tab. 5 indicate that the prompt ordering (ID-Scene vs. Scene-ID) has a non-negligible impact on the metrics, particularly for the baseline model. We hypothesize that tokens placed earlier in the prompt receive more conditioning weight in the cross-attention mechanism, driving the observed sensitivity:

Table 5: Effect of scene-ID prompt order.

| Method | DreamSim-F↓ | CLIP-I↑ | DreamSim-B↑ | CLIP-T↑ |
|--------|-------------|---------|-------------|---------|
| SDXL-o | 0.2778 | 0.8558 | 0.3861 | 0.8865 |
| SDXL-r | 0.2894 (+0.0116) | 0.8507 | 0.3966 (+0.0105) | 0.8899 |
| SDeC-o | 0.2589 | 0.8655 | 0.3675 | 0.8946 |
| SDeC-r | 0.2670 (+0.0081) | 0.8609 | 0.3777 (+0.0102) | 0.8935 |

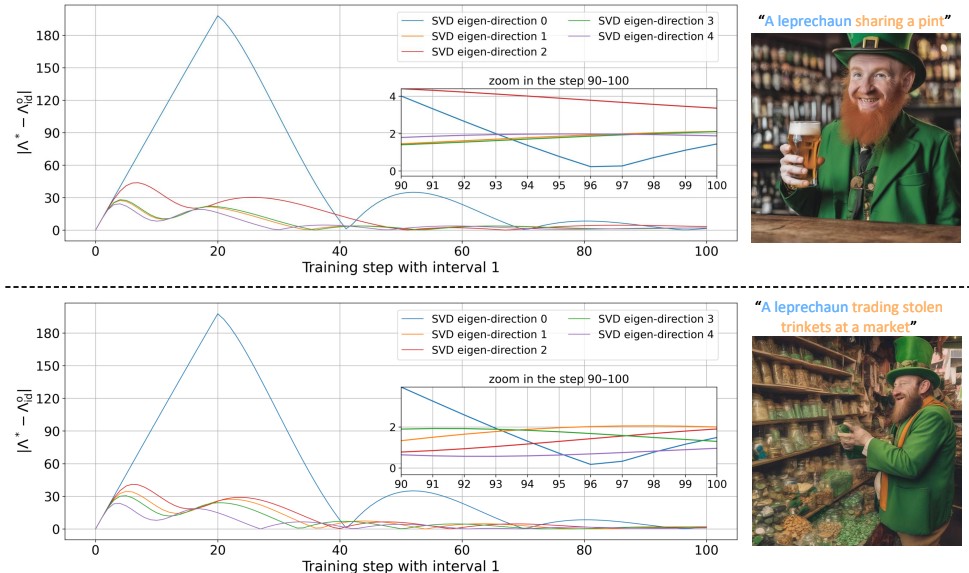

Figure 11: SVD eigenvalues evolving dynamics relative to their original values during the "forward-and-backward" optimization. Here, in the "SVD eigen-direction **#**," a smaller value of **#** indicates a larger eigenvalue for that direction.

1. *Baseline (SDXL) behavior*: When the prompt order is reversed (Scene-ID), the baseline model shows worse ID consistency (higher DreamSim-F score) but better scene distinction (higher DreamSim-B score). This suggests the scene tokens, when placed earlier, exert a stronger, negative influence on the subject's identity, while simultaneously making the scene more distinct.

2. *SDeC behavior*: When SDeC is applied, the negative effect on the ID from prompt reversing is slightly reduced, which is consistent with our objective of suppressing scene-ID correlation. Importantly, the positive effect on scene distinction is maintained, as SDeC leverages the original scene prompt content.

### H.4 ANALYSIS OF THE "FORWARD-AND-BACKWARD" OPTIMIZATION

In SDeC, the "forward-and-backward" optimization is central to identifying the latent scene–ID correlation subspace within the ID prompt. To better understand this process, we visualize how the SVD eigenvalues evolve relative to their original values during optimization. For clarity, we show an example with five ID tokens, resulting in five corresponding SVD eigen-directions. We present the visualization results in Fig. 11. There are three observations below.

First, the evolution of the SVD eigenvalue gap against the original value follows our expectation. In the forward phase (step 0-20), the eigenvalue distinction curves gradually grow to pull the ID embedding to the scene embedding. In the subsequent backward phase (step 20-100), despite some vibration (e.g., the blue one), the curves globally converge to zero. This verifies the recovery of the ID-associated component.

Second, in the final step, the eigenvalue gaps exhibit a clear divergence, which allows us to treat the directions with larger gaps as the basis of the ID–scene shared subspace. Moreover, this divergence is

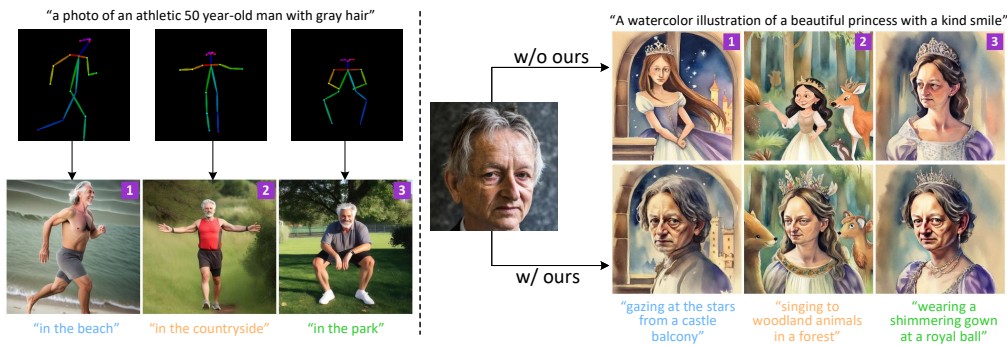

Figure 12: **Left**: Incorporating SDeC with ControlNet under the control of pose map. **Right**: Integrating SDeC with PhotoMaker, where a photo of *Geoffrey Hinton* serves as the reference.

correlated with the specific scene prompt. For example, in the top case, the gap is most pronounced along eigen-direction 2 (red), whereas in the bottom case, the divergence appears along eigen-directions 1 and 2. This also indicates that the "forward-and-backward" optimization imposes a scene-specific de-contextualization.

Third, the eigen-directions with mid-range eigenvalues, such as eigen-directions 1 and 2, exhibit larger final gap values and are thus more readily identified as the basis of the ID–scene shared subspace. This behavior naturally emerges from the "forward-and-backward" optimization: compressing the mid-eigenvalue directions simultaneously preserves the dominant ID-embedding information encoded in the largest-eigenvalue directions while maintaining sufficient compression strength to meaningfully influence the generation process.

# I SUPPLEMENTAL EXPERIMENTS

## I.1 INTEGRATING WITH OTHER GENERATIVE TASKS

SDeC reduces the ID shift by editing the ID prompt embedding, without modifying the generative models. Consequently, it is compatible with different visual generation tasks. In this part, we incorporate the proposed method with two typical models with different generative goals. One is ControlNet (Zhang et al., 2023), which introduces controllable conditions (e.g., edge maps, pose maps, or depth maps) to enable structured control over image generation. The other is PhotoMaker (Li et al., 2024), which leverages an input reference image to preserve identity features, generating consistent subject images across diverse scenarios. As shown in Fig. 12, our SDeC demonstrates excellent compatibility and ID-preservation.

## I.2 CONSISTENT STORY GENERATION WITH MULTIPLE SUBJECTS

In this part, we present the effect of SDeC as the ID prompt involves multiple subjects. Fig. 13 demonstrates a toy case involving an "elderly man" and a "cat". It can be observed that these two subjects are basically consistent, although some ID shift: The shorter cat fur in images #4, #9, and #10; the absence of glasses in images #1, #7; the hat in image #8.

## I.3 GENERALITY TO BASE GENERATIVE MODEL

Extensive experiments with the SDXL base model show that our SDeC enables semantically meaningful editing. However, if its effectiveness were confined to only a subset of generative models, its practical applicability would be greatly limited. To demonstrate its generality, we integrate SDeC with four representative base models and compare performance before and after integration. Specif-

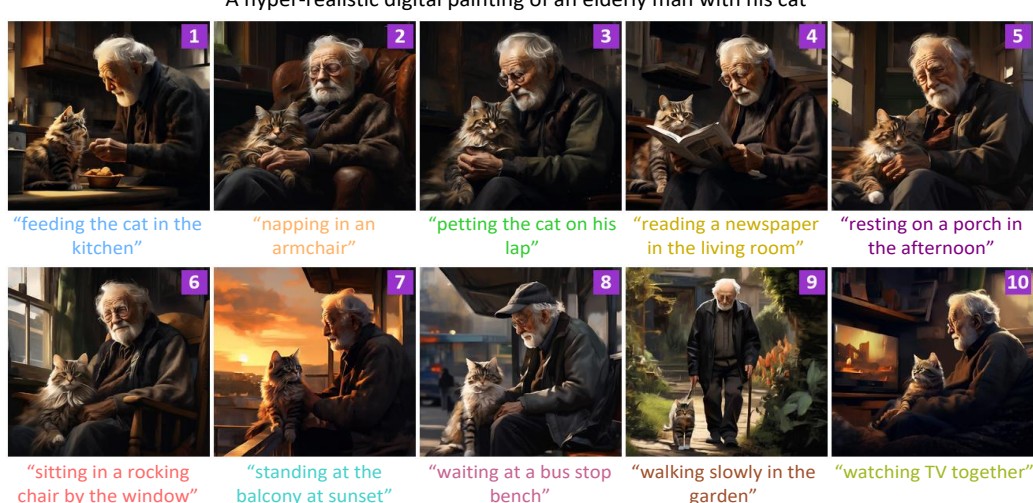

Figure 13: Consistent story generation with multiple subjects. The results show that our SDeC can generate images featuring multiple expected characters, with minor ID shift, for example, the hat (image #8) and glasses (images #1, #7).

Table 6: Quantitative results obtained by combining SDeC with four UNet-based generative base models (**Top**) and two MMDiT-based generative base models (**Bottom**).

| Base-model type | Method | DreamSim-F↓ | CLIP-I↑ | DreamSim-B↑ | CLIP-T↑ |
|---|---|---|---|---|---|
| UNet | SDXL | 0.2778 | 0.8558 | 0.3861 | 0.8865 |
| | SDXL+**SDeC** | 0.2589 | 0.8655 | 0.3675 | 0.8946 |
| | PlayGround-v2.5 | 0.2567 | 0.8680 | 0.3688 | 0.8799 |
| | PlayGround-v2.5+**SDeC** | 0.2272 | 0.8832 | 0.3470 | 0.8994 |
| | RealVisXL-V4.0 | 0.2616 | 0.8660 | 0.4075 | 0.9007 |
| | RealVisXL-V4.0+**SDeC** | 0.2435 | 0.8665 | 0.3845 | 0.8942 |
| | Juggernaut-X-V10 | 0.2748 | 0.8572 | 0.4295 | 0.8974 |
| | Juggernaut-X-V10+**SDeC** | 0.2369 | 0.8781 | 0.4011 | 0.9077 |
| MMDiT | SD3 | 0.2875 | 0.8495 | 0.4236 | 0.8596 |
| | SD3+**SDeC** | 0.2644 | 0.8612 | 0.4140 | 0.8629 |
| | Flux | 0.2844 | 0.8516 | 0.4238 | 0.8632 |
| | Flux+**SDeC** | 0.2653 | 0.8623 | 0.4167 | 0.8648 |

ically, in addition to SDXL, the other base models include PlayGround-v2.5-1024px-Aesthetic[4], RealVisXL-V4.0[5], and Juggernaut-X-V10[6].

From the comparison presented in Fig. 14, we draw two main observations. First, within the SDXL group, SDeC substantially alters the subject in image #2, while making only minor adjustments to the others, for example, consistently sharpening the cats' chins. This is reasonable, as SDXL already performs well in ID preservation for the remaining images. Second, in contrast, when the base models exhibit evident identity divergence (see the other three groups), equipping them with SDeC leads to a significant improvement in the quality of the generated images.

The base models above are based on the UNet (Ronneberger et al., 2015) architecture. To verify the generality of our method, we further evaluate our approach using two generative models based on MMDiT (Esser et al., 2024): SD3[7] and Flux[8]. As shown in Fig. 15, in the SD3 group, the model equipped with our SDeC achieves substantial improvements in appearance consistency, including

---

[4]https://huggingface.co/playgroundai/playground-v2.5-1024px-aesthetic
[5]https://huggingface.co/SG161222/RealVisXL_V4.0
[6]https://huggingface.co/RunDiffusion/Juggernaut-X-v10
[7]https://huggingface.co/stabilityai/stable-diffusion-3-medium
[8]https://huggingface.co/black-forest-labs/FLUX.1-dev

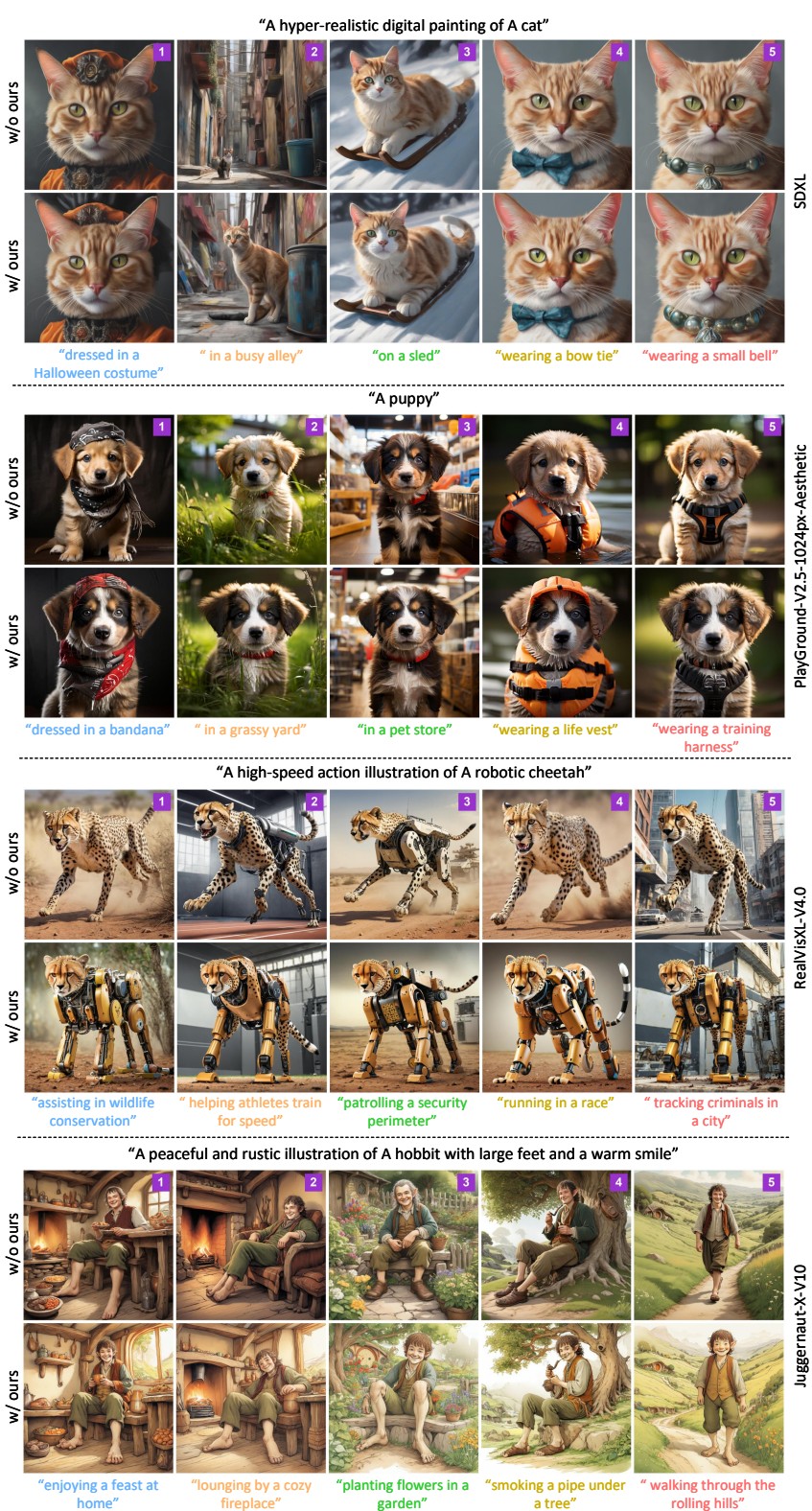

Figure 14: Comparison of combining SDeC with the UNet-based generative models. From **top** to **bottom**, there are results of **SDXL** group, **PlayGround-v2.5-1024px-Aesthetic** group, **RealVisXL-V4.0** group, and **Juggernaut-X-V10** group, respectively.

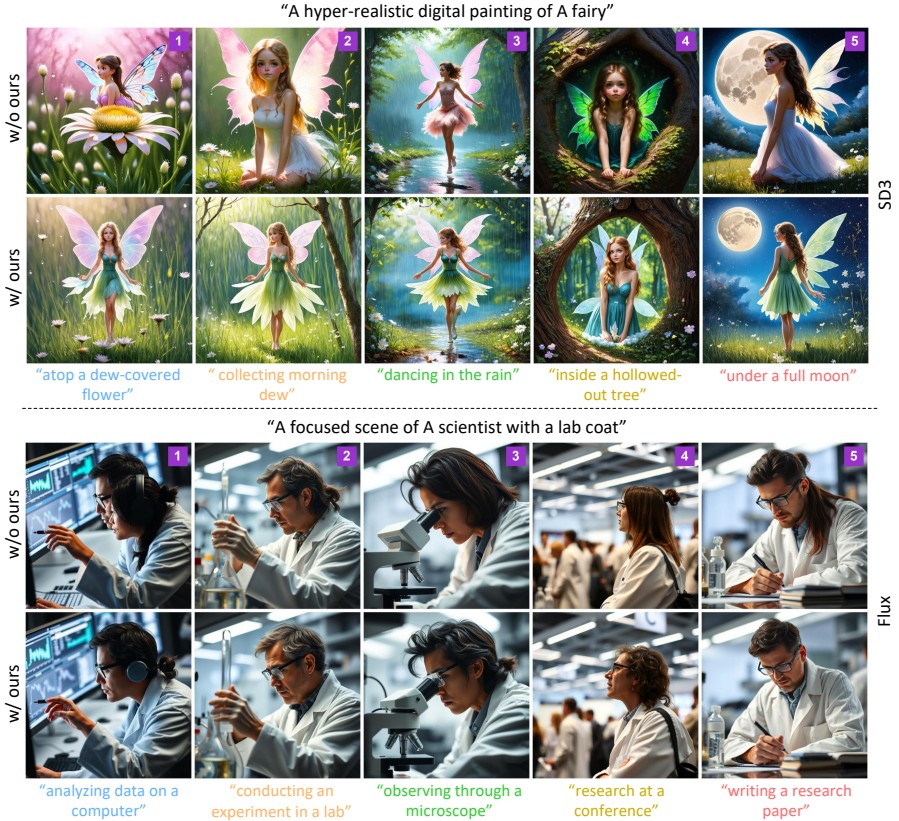

Figure 15: Comparison of combining SDeC with the MMDiT-based generative models. **Top**: Results of **SD3** group. **Bottom**: Results of **Flux** group.

clothing, hair color, posture, and wing features. In the Flux group, the original Flux model shows an ID shift in gender, whereas incorporating our method significantly mitigates this issue.

For a comprehensive comparison, we list the quantitative results of the methods mentioned above in Tab. 6. In each group, the base model equipped with SDeC achieves stronger ID-related performance (DreamSim-F, CLIP-I) and comparable scenario-level results (DreamSim-B, CLIP-T) relative to the original base models. These results confirm SDeC's effectiveness and architectural independence.

## J AN EMPIRICAL STUDY OF NANO BANANA'S ID-PRESERVATION

### J.1 BUILDING BASELINE

As the beginning of our exploration, this baseline experiment generates 12 images using Nano Banana[9] under the setting formulated in the `Problem Statement` of Sec. 3. In practice, we start a new session and sequentially input the 12 prompts into Nano Banana via its official interface in dialogue mode. We set the ID prompt to "*A dreamy illustration of a beautiful princess with a kind smile*", and the scene prompts are generated using ChatGPT-5.0[10].

As shown in Fig. 16, the generated images exhibit excellent ID consistency, except for the hairstyle in image #4 and the clothing in image #5. We further observe that the generation time increases steadily from **8.5** seconds for the first image to **82.6** seconds for the last. This phenomenon suggests that Nano Banana might leverage previously generated images as contextual information during subsequent generation. If this speculation holds, Google's method implicitly incorporates a reference image to

---

[9] https://aistudio.google.com/models/gemini-2-5-flash-image
[10] https://chatgpt.com/

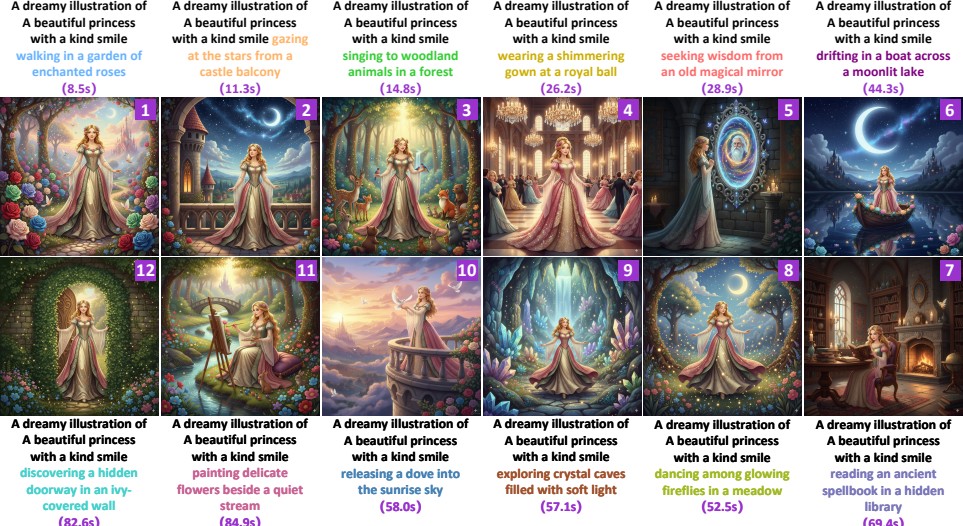

Figure 16: Results of baseline experiment where the images are arranged in a U-shape. All scenarios share the same ID prompt, and the scene prompts are different, as in the setting of this paper.

enforce ID consistency, the same as the personalized T2I generation (see `Sec.2 Related Work`). Accordingly, the strong ID preservation becomes interpretable. In Nano Banana, the developed image editing technique (a feature also emphasized officially) is employed to integrate the subject from the reference image with the target scene. The subjects in images #1, #2, #4, #6, #8, and #9 provide compelling support: Their clothing and pose are mirrored, which exhibits clear signs of editing.

## J.2 VALIDATING LEVERAGING PRIOR IMAGES AS CONTEXT

As presented above, our conclusion regarding reference images is drawn under the assumption that contextual information is exploited in Nano Banana. To test this assumption, we selected scenario #1 and #4 from the baseline as the first and last ones, respectively, and inserted 10 intermediate scenarios between them. Based on this, we conducted two perturbation experiments as follows. Of note, to avoid interference between experiments, each experiment is conducted in a newly created session.

In the first experiment, we introduce substantial perturbations to the intermediate scenes by asking ChatGPT-5.0 to produce prompts that are markedly different from the first scenario. The results in Fig. 17 reveal significant differences between the first and last images (e.g., hairstyle, costume, and headdress), suggesting that the Nano Banana model indeed accounts for contextual information.

In the second experiment, we perform minor perturbations: The intermediate scenarios adopt a literary style similar to that of the first scenario, achieved by instructing ChatGPT-5.0 to replicate its style literally. Meanwhile, the ID component remains aligned with the first, but is slightly varied by substituting the subject with related terms (e.g., "woman" and "girl"). As shown in Fig. 18, compared with the baseline, the subjects in the first and last images again exhibited notable differences (e.g., hair color, hair length, and costume style). Moreover, the last image and its neighbors displayed higher facial similarity than the first. We further extend the number of scenarios to Nano Banana's maximum support (**22**). The results in Fig. 19 reveal that the differences between the first and last images are further amplified, making it difficult to regard the presented subjects as the same person.

While the observed ID shifts under the two types of perturbations suggest Nano Banana adopts a context strategy, another evidence arises from the side-by-side comparison of experimental results. Specifically, compared with minor perturbations (Fig. 18 and Fig. 19), strong perturbations (Fig. 17) result in smaller ID shifts. A reasonable explanation is that in the strong-perturbation case, the constructed context exhibits much larger semantic differences. This enables the generation process of image #12 to more easily converge attention on the most similar first scenario, thereby achieving ID-preservation. In contrast, in the minor-perturbation case, the attention distribution remains relatively

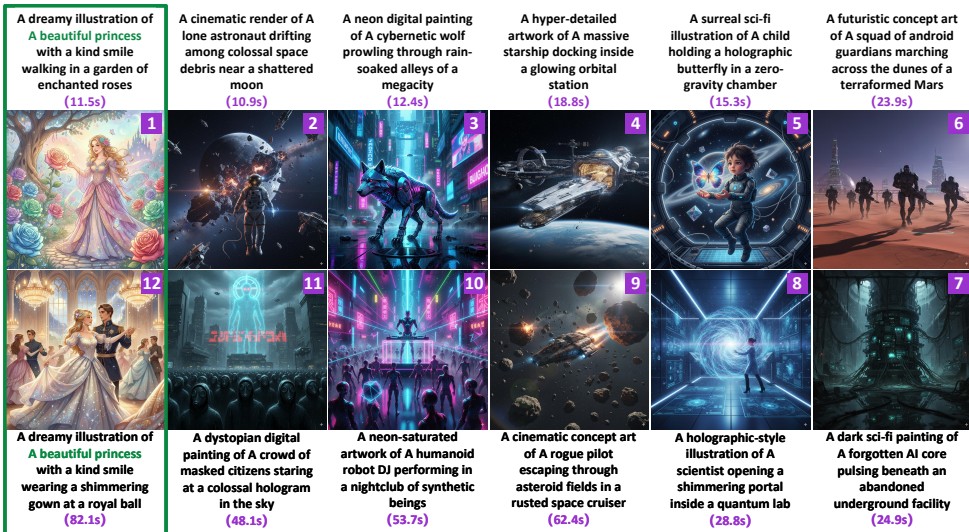

Figure 17: Results of experiment with major perturbation where the images are arranged in a U-shape. The intermediate scenarios (#2~#11) significantly differ from the first one.

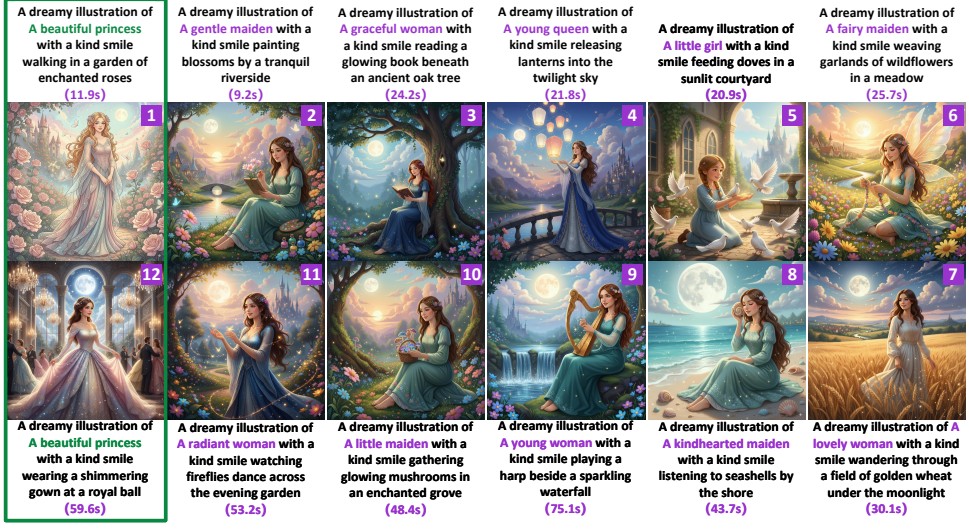

Figure 18: Experiment with minor perturbation where the images are arranged in a U-shape. In the intermediate scenarios (#2~#11), the scene prompts generated by ChatGPT-5.0 follow a literary style similar to that of the first, while the ID prompts remain consistent with the first, differing only in the substitution of the subject with a related concept.

flat. As a result of blending the semantics of multiple scenarios, the final generated image displays a much more pronounced ID shift.

## J.3 COMPARING WITH OUR METHOD IN THE WAY OF ID-PRESERVATION

In summary, the ID consistency observed in Nano Banana can be attributed to the implicit use of reference images, where previously generated results serve as contextual input. This allows image editing techniques to enforce high-quality ID consistency. In contrast, our method avoids such strong assumptions: The "one prompt per scene" feature provides an unconstrained usage condition and requires neither intensive computation nor additional data overhead.

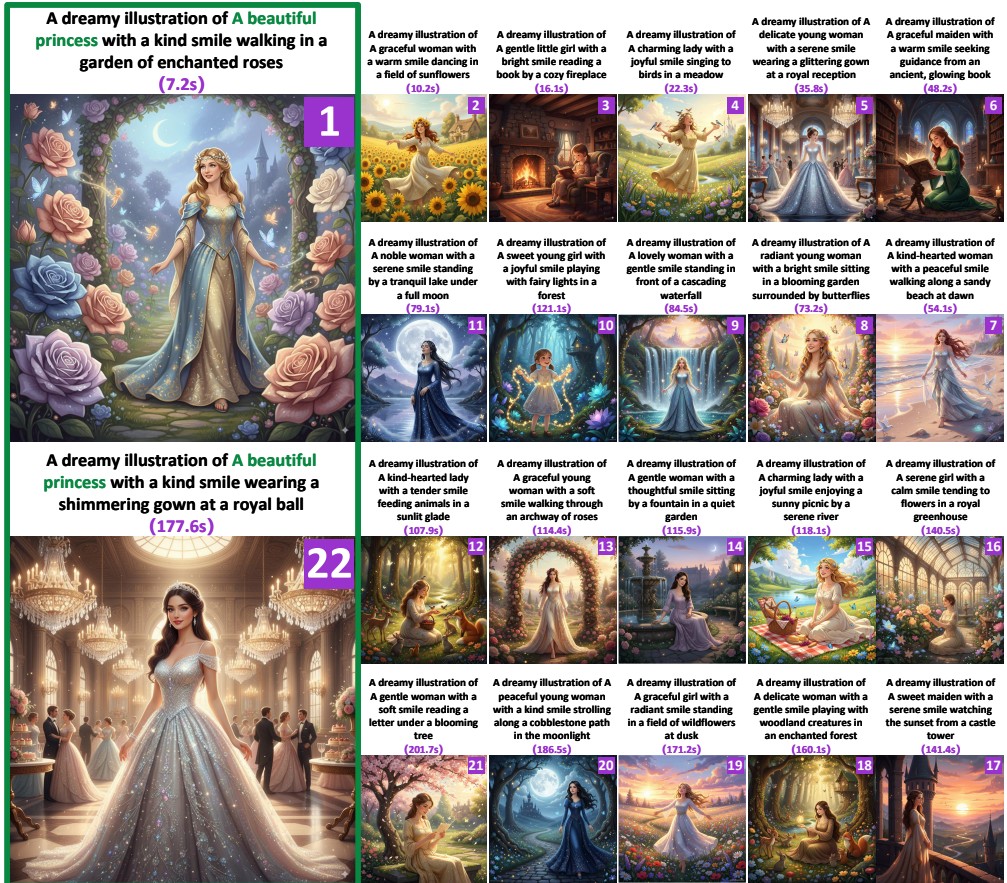

Figure 19: Extended experiment with minor perturbations, under the same setting as Fig. 18, with the number of images increased to the maximum continuous generation times of Nano Banana.

Methodologically, Nano Banana falls within the category of personalized T2I generation. It typically relies on a reference dataset, taken as context, to model ID invariance, aligning with the principles of transfer learning. In contrast, SDeC takes a conceptually distinct path by pursuing a novel prompt embedding editing paradigm, derived from a native generative perspective on ID shift: Scene contextualization in each individual image.

### J.4 COMPARING WITH OUR METHOD UNDER SCENE WITH STRONG VISUAL SHIFTS

The de-contextualization studied in this paper focuses on the general effects of a scene on identity (e.g., changes in clothing or posture due to a scene description), which covers normal usage scenarios. If the scene is an extreme case involving strong visual shifts (e.g., intense color washes that fundamentally alter the subject's appearance, far beyond normal shadow changes), would our method still remain effective?

To address this question, we present the outputs of our SDeC combined with the SDXL base model (denoted by SDXL+SDeC), where the scene prompt includes both normal- and strong-visual shifts. For comparison, we also include results generated by Nano Banana.

From Fig. 20, we observe that the newly released commercial model Nano Banana and SDXL+SDeC fail to capture the direction of the light source, which is crucial for determining shadow formation. This indicates that their ability to interpret and realize the strong visual shifts specified in the scene prompt remains limited.

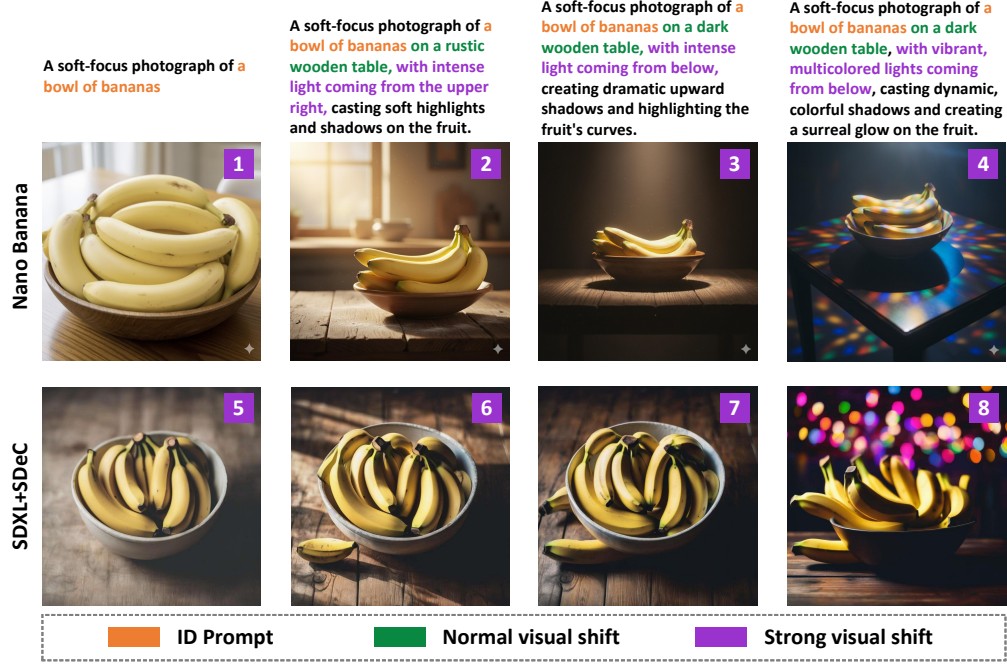

Figure 20: Comparison results under scene with strong visual shifts.

## K    LIMITATION AND FUTURE WORK

Despite improving ID consistency while maintaining scene diversity, SDeC, as a prompt-embedding editing approach, cannot fundamentally resolve ID shift. First, Theorem 1 proves that the attention mechanism is the central origin of ID shift, even when ID and scene prompt embeddings occupy disjoint subspaces. Second, in line with Theorem 2, SDeC essentially performs an indirect form of contextualization control by reducing the overlapping energy between ID and scene embeddings.

Therefore, designing attention modules tailored to preserve ID represents a promising direction for future work. In this paper, however, our theoretical contributions, such as Theorem 1 and Corollary 1, are intended to reveal the mechanistic link between scene contextualization and ID shift. Their practical contribution to attention design remains limited. This limitation arises from idealized assumptions (e.g., sharp subspace partitioning and linearity) and the neglect of real data geometry and attention dynamics. These factors render the precise construction of $P_\cap$ numerically unstable and difficult to apply in high-dimensional, sample-limited settings. Addressing these challenges will be a central focus for future work.

Additionally, a key motivation of this work is the lack of theoretical justification for ID shift in the T2I literature. We provide the first theoretical interpretation of ID shift through the lens of scene contextualization, and offer a practical solution by addressing the underlying correlation. Nonetheless, this represents only the beginning of a deeper exploration into the fundamental mathematical "balance" between identity and scene. In this paper, we observe two pieces of evidence suggesting the existence of a possible balance. One is the trade-off visualized in Fig. 10 (Middle). The other one is the disentanglement-coherence trade-off reflected in Corollary 2. Developing a systematic theoretical framework for this balance constitutes an important and promising direction for future research.

At present, addressing these extreme visual alterations remains a significant challenge for all current generative models. The results in Fig. 20 highlight the limitations of our method in this regard. Consequently, explicitly incorporating the constraints imposed by extreme visual alterations into the prompt-editing process constitutes an interesting direction for future research.

