# OpenReview forum: "Consistent Text-to-Image Generation via Scene De-Contextualization"
_ICLR.cc/2026/Conference — ICLR 2026 Poster_

### Official Review · Reviewer_Hj37 · 2025-10-19

**Soundness:** 3
**Presentation:** 3
**Contribution:** 3
**Rating:** 6
**Confidence:** 4

**Summary:**

This paper aims to prevent ID drift in text-to-image based personalized storytelling. This paper theoretically analyze the cause of ID shift, which is named scene contextualization. Based on the theory, this paper further quantifies the contextualization, and proposes scene de-contextualization. The idea is to quantify the extent to which each direction is influenced by contextualization and then selectively reinforce those that are less affected.

In practice, it first pull the eigenvalue of SVD of ID embedding towards scene, and then pull the eigenvalue back towards the ID. The directions whose corresponding eigenvalues remain nearly unchanged (resistant to both pull and restoration) are treated as robust directions against contextualization. In contrast, those with large variations are the latent scene-ID correlation subspace. To suppress the scene influence on ID, it suppresses eigenvalue with large variations by using adaptive weighting. The modified ID embedding are fed into the model along with the original scene embedding.

Experiments show that this method can be a plug-in into diverse tasks, e.g., integrating pose map and personalized photo, and generative backbones such as PlayGround-v2.5, RealVisXL-V4.0 and Juggernaut-X-V10.

**Strengths:**

1. This paper seems to strike a new balance between ID consistency and scene diversity.
2. This paper derives the method from theory, which may inspire future researchers.

**Weaknesses:**

1. Does it really make sense to suppress $\sigma_{\cap}$? What if the scene has very strong information on the subject such as colorful lighting?
2. Related to 1), the method seems sensitive to $\Omega$ as shown in Figure 8. As both works on SVD eigenvalues, this paper seems an improvement on 1Prompt1Story. In this sense, sensitivity to  $\Omega$ can significantly harm the contribution.

**Questions:**

1. Maybe more math: what might be the (possibly) deepest mathematical reason of there being always a balance between ID and scene? Since this paper already contains a lot of math, digging into the reason can make this paper in a higher stage.
2. Maybe more intuition: I feel visualizing more about the pull and restoration process and playing with $\Omega$ might give some more insight.

---

> ### Author Response · Authors · 2025-11-22
>
> **Q1**: Does it really make sense to suppress $\sigma_\cap$ ? What if the scene has very strong information on the subject such as colorful lighting?
>
> **R**: Insightful perspective.We agree that suppressing the scene-ID correlation ($\sigma_\cap$) might be challenged by extreme contextual changes, like colorful lighting.
>
> Our current research focuses on the **general effects of scene on identity** (e.g., changes in clothing or posture due to a scene description), which covers **normal usage scenarios**.
>
> The "colorful lighting" example is an **extreme case** involving strong visual shifts (e.g., intense color washes that fundamentally alter the subject's appearance, far beyond normal shadow changes). Addressing these extreme visual alterations remains a significant challenge for all current generative models (Appdendix-J.4). These extreme cases present more challenges and require dedicated research in the future. We have discussed this important challenge in the **Limitations and Future Work section** (Appendix-K).
>
>
> **Q2**: The method seems sensitive to as shown in Figure 8. As both works on SVD eigenvalues, this paper seems an improvement on 1Prompt1Story. In this sense, sensitivity to can significantly harm the contribution.
>
> **R**: Regarding the sensitivity of the performance to the $\Omega$ and $\beta_M$ parameters (Figure 10):
>
> 1.  **Sensitivity as evidence:** We intentionally used large parameter intervals to emphasize the performance variation. The observed sensitivity confirms that our editing mechanism is actively working by controlling the trade-off between ID consistency and scene preservation.
> 2.  **Robust operating range:** Crucially, our analysis confirms a *reasonable wide, stable range of values* exists for both parameters that yields strong performance. This ensures straightforward, non-fragile setting for robust, easy deployment.
>
> Despite both SDeC and 1Prompt1Story (1P1S) utilizing SVD eigenvalues, their *ideas and technical approaches are fundamentally different*:
>
> * **Objective:** 1P1S aligns scenes in an integrated prompt. SDeC diagnoses the problem by identifying and suppressing the latent scene-ID correlation subspace **for each individual scene**.
> * **Technique:** 1P1S uses simple eigenvalue suppression for alignment. SDeC employs a more sophisticated mechanism: it quantifies the robustness of each eigen-direction by examining the variation of SVD eigenvalues during a "forward-and-backward" optimization process to reliably identify the scene–ID correlation subspace.
>
>
> **Q3**: Maybe more math: what might be the (possibly) deepest mathematical reason of there being always a balance between ID and scene? Since this paper already contains a lot of math, digging into the reason can make this paper in a higher stage.
>
> **R**: Thanks for this highly insightful comment, which points toward a crucial area for future theoretical work.
>
> One of the major motivations for this work is exactly the *lack of theoretical justification* for ID shift in the T2I literature. This work provides the **first theoretical interpretation of ID shift** through scene contextualization (Theorem 1 and Corollary 1).
>
> While our work addresses the correlation and provides a solution, we recognize this is just the **start** of exploring the problem of the deeper mathematical "balance" between ID and scene. Figure 10 (Middle) already illustrates this necessary trade-off.
>
> A systematic, deeper mathematical study of this "balance" is a promising theoretical direction as highlighted in the revised **Limitations and Future Work section** (Appendix-K).
>
>
> **Q4**: Maybe more intuition: I feel visualizing more about the pull and restoration process and playing with $\Omega$ might give some more insight.
>
> **R**: Excellent point! To aid understanding of the pull-and-restoration process, we have further elaborated on our motivation:
>
> *"In the two-phase optimization defined in Eq. (6), the forward phase identifies directions in $Z_{\mathrm{id}}$ that align with $Z_{\mathrm{sc}}^{k}$, capturing their shared representations. However, some of these directions may also be essential for representing $Z_{\mathrm{id}}$ itself. To mitigate potential semantic degradation in $Z_{\mathrm{id}}$, we incorporate a backward phase that progressively restores these ID-associated components."*
>
>
>
> Additionally, we have visualized the eigenvalues' variation with respect to the original values during this pull and restoration process (**see Appendix-H.5** in the revised revision for the results and analysis).
>
> Regarding the parameter $\Omega$, **Fig. 10 (Middle)** already shows that it affects the balance between ID and scene. This may provide a promising entry point for further exploring the balance issue as raised. We have discussed this in the **Limitations and Future Work section** (Appendix-K).

---

> > ### Comment · Reviewer_Hj37 · 2025-11-25
> >
> > Thanks for the response. My concern is partially solved. The unsolved part has been added to Limitations. Since I have originally gave 6, I keep my score.

---

> ### Author Response · Authors · 2025-11-26
>
> Thanks for the timely response. For the new comment, we provide our reply as follows.
>
> **C1**. My concern is partially solved. The unsolved part has been added to Limitations.
>
> **R**: In our previous response, we have already provided **substantive answers** to all reviewer comments. The issue mentioned by the reviewer, **“the unsolved part has been added to Limitations,”** concerns Q1, Q3, and Q4. The corresponding revisions are as follows:
>
> 1. **Q1**: In the newly added **Appendix J.4**, we have included new experiments showing that, under extreme visual-shift scenarios, our method still has limitations. Notably, the leading commercial model **Nano Banana** also fails in these extreme cases, suggesting that this remains **an open challenge**.  We therefore left it as future work.
>
> $ $
>
> 2. **Q3**: We discussed the **ID–scene trade-off** shown in Fig. 10 (Middle) in **Lines 1041–1047** and clarified the **disentanglement–coherence balance** reflected in Corollary 2 (Eq. (5)) in **Lines 231–235**. Although the analyses suggest a possible balance, developing a rigorous mathematical formulation would require initiating a new line of research, which is **beyond the scope of this paper**. Therefore, we list this as future work.
>
> $ $
>
> 3. **Q4**: In addition to the $\Omega$-induced D–scene trade-off shown in Fig. 10 (Middle), we expanded the **explanation of the empirical selection of hyperparameters**, including the role of $\Omega$, in **Lines 1022–1036** of the revised version.
>
>
> We believe these revisions address all raised concerns. If there are any points we may have missed, or if further clarification would be helpful, we would greatly appreciate the reviewer’s feedback.

---

> > ### Comment · Reviewer_Hj37 · 2025-11-27
> >
> > I appreciate the quick reply. I think that solves all my concerns, and I do believe confidently showing the limitation (via visualization) makes the paper more beneficial to the community. I have increased my rating to 8.

---

> > > ### Author Response · Authors · 2025-11-28
> > >
> > > Thank you for your feedback and consideration!

---

### Official Review · Reviewer_GF6P · 2025-10-25

**Soundness:** 3
**Presentation:** 3
**Contribution:** 3
**Rating:** 6
**Confidence:** 2

**Summary:**

This paper investigates the cause of identity (ID) shift in text-to-image (T2I) models and introduces a training-free method called Scene De-Contextualization (SDeC). The authors theorize that ID shift arises from *scene contextualization*—an intrinsic correlation between identity and scene semantics induced by the attention mechanism. They formally prove this phenomenon’s inevitability and bound its strength. Building on this, SDeC edits prompt embeddings to suppress latent scene-ID correlations by analyzing singular value stability and adaptively reweighting eigenvalues, allowing per-scene use without knowing all target scenes. Experiments on ConsiStory+ demonstrate that SDeC improves identity consistency while preserving scene diversity and works complementarily with prior methods like ConsiStory. The paper’s theoretical analysis and efficient, training-free implementation make it a notable step toward understanding and mitigating ID shift in T2I generation.

**Strengths:**

1. The paper provides a clear and rigorous formulation of *scene contextualization* as the root cause of identity shift in T2I models, supported by formal theorems and bounds. This theoretical framing is original and intellectually valuable.
2. The proposed SDeC method operates purely at the prompt-embedding level, requiring no model retraining or access to all target scenes, which makes it lightweight, general, and practical.
3. The paper is well structured, linking theoretical findings to practical implications, and provides convincing visual and analytical evidence for the proposed claims.

**Weaknesses:**

1. The Introduction could be made clearer for readers unfamiliar with this research area. For example, in *line 47*, the authors argue that “target scenes are not always available,” but it is unclear whether this refers to the incompleteness of scene samples in the training data. When introducing *scene contextualization*, it would also help to include a comparative visualization in Figure 1 showing scenes with and without ID shift, to better illustrate the motivation.
2. The theoretical analysis in Section 3 is difficult to follow, especially for non-expert readers. From a high-level understanding (as shown in *Figure 2*), *scene contextualization* arises because the text encoder’s causal attention mechanism allows scene tokens to capture partial semantics from ID tokens, thereby causing ID shift. However, I have two questions: (i) does this phenomenon only occur in CLIP-based T2I models, since newer architectures such as SD3and Flux use T5-based encoders with *bidirectional attention*? Would the proposed method still apply under such architectures? (ii) How sensitive is this effect to the order of the scene and ID prompts? For instance, if the scene prompt precedes the ID prompt, would scene contextualization still emerge, and would SDeC remain effective in that case?
3. As I am not fully up to date with the latest progress in this field, I am unable to fully assess the choice of baselines and the reported performance comparisons, and therefore defer to other reviewers for judgment on that aspect.

**Questions:**

See the weakness part.

---

> ### Author Response · Authors · 2025-11-22
>
> **Q1**: The Introduction could be made clearer for readers unfamiliar with this research area. For example, in line 47, the authors argue that “target scenes are not always available,” but it is unclear whether this refers to the incompleteness of scene samples in the training data. When introducing scene contextualization, it would also help to include a comparative visualization in Figure 1 showing scenes with and without ID shift, to better illustrate the motivation.
>
> **R**: Great suggestions!
>
> 1.  **Target scene availability (Line 46):** The argument that "target scenes are not always available" refers to the practical, dynamic nature of project development, not training data incompleteness. In real-world projects (e.g., films, games, or story creation), the full set of final scenes, their content, and their order are often refined and finalized over numerous iterative changes, making it impossible to know all subsequent scene contexts in advance. Efficiency dictates generating images online based on the current scene description, **without the need for repeatedly re-generating all previous scenes (avoiding exponential complexity)**. We have clarified this practical setting in the revised Introduction (Lines **051-053**).
>
>
> 2.  **Visualizing no ID Shift (Figure 1):** As ID shift naturally arises under standard generation, making a truly shift-free counter-example difficult to isolate for pure visualization.
>
>
> **Q2**: Does this phenomenon only occur in CLIP-based T2I models, since newer architectures such as SD3 and Flux use T5-based encoders with bidirectional attention? Would the proposed method still apply under such architectures?
>
> **R**: As estalbished in our Theorem 1 and Corollary 1, the **attention architecture inevitably induces scene contextualization**, regardless of whether the identity and scene semantic subspaces overlap. As a particular form of attention, the bidirectional attention used in SD3 and Flux likewise leads to scene contextualization. Therefore, our theory is expected to remain valid under these architectures.
>
> To verify this, we further evaluated our approach on two DiT-based generative models, SD3 and Flux. As shown in our response to **Q1** of reviewer **xuex**, the results confirm that the proposed method consistently enhances these newer architectures.
>
>
> **Q3**: How sensitive is this effect to the order of the scene and ID prompts? For instance, if the scene prompt precedes the ID prompt, would scene contextualization still emerge, and would SDeC remain effective in that case?
>
> **R**: Great thought! To investigate the effect of scene-ID order, we conducted an additional experiment with scene and ID prompts swapped. The Table below shows that the prompt order makes **marginal impact** (**see Appendix-H.4** in the revised paper)
>
>
> Tab. Effect of scene-ID prmot order. "-r": reversing ID and scene prompt.
> |Method|DreamSim-F↓|CLIP-I↑|DreamSim-B↑|CLIP-T↑|
> |-|-|-|-|-|
> |SDXL    |0.2778|0.8558|0.3861|0.8865|
> |SDXL-r  |0.2894|0.8507|0.3966|0.8899|
> |SDeC    |0.2589|0.8655|0.3675|0.8946|
> |SDeC-r  |0.2670|0.8609|0.3777|0.8935|
>
>
> **Q4**: As I am not fully up to date with the latest progress in this field, I am unable to fully assess the choice of baselines and the reported performance comparisons, and therefore defer to other reviewers for judgment on that aspect.
>
> **R**: Thanks! As a recap, we cover all recent state-of-the-art methods in consistent T2I generation: BLIP-Diffusion was published in **2023**; PhotoMaker, ConsiStory, and StoryDiffusion appeared in **2024**; and 1Prompt1Story is a **2025** work.
>
> Beyond these academic works, we also included an empirical comparison with the Google's latest commercial product **Nano Banana** (**Appendix-J**), which went alive on August 26, *only one month before the ICLR 2026 submission deadline*.
>
> We believe this work sufficiently reflects the latest advances in consistent T2I generation and provides a fair and up-to-date evaluation.

---

> > ### Comment · Reviewer_GF6P · 2025-11-27
> >
> > Thanks for the authors' response. Most of my concerns have been addressed. Regarding the scene-ID order in Q3, from the quantitative results it appears that the ordering has a non-negligible impact on the metrics. Could the authors provide some analysis or insights into why this happens and what underlying factors might be driving the sensitivity to ordering?

---

> > > ### Author Response · Authors · 2025-11-28
> > >
> > > **R**: Thanks for timely response and insightful suggestion. We agree that the prompt ordering (ID-Scene vs. Scene-ID) has a non-negligible impact on the metrics, particularly for the baseline model.
> > >
> > > We hypothesize that **tokens placed earlier in the prompt receive more conditioning weight in the cross-attention mechanism**, driving the observed sensitivity:
> > > 1. **Baseline (SDXL) behavior**: When the prompt order is reversed (Scene-ID), the baseline model shows worse ID consistency (higher DreamSim-F score) but better scene distinction (higher DreamSim-B score). This suggests the scene tokens, when placed earlier, exert a stronger, negative influence on the subject's identity, while simultaneously making the scene more distinct.
> > >
> > > $ $
> > >
> > > 2. **SDeC behavior**: When SDeC is applied, the negative effect on the ID from prompt reversing is slightly reduced, which is consistent with our objective of suppressing scene-ID correlation. Importantly, the positive effect on scene distinction is maintained, as SDeC leverages the original scene prompt content.
> > >
> > > We have now added this detailed analysis to the revised paper (Section **H.4** in Appendix).
> > >
> > > $ $
> > >
> > > Tab. Effect of prompt order. "-o": ID-Scene (original order), "-r": Scene-ID (reversed).
> > > |#|Method|DreamSim-F↓|CLIP-I↑|DreamSim-B↑|CLIP-T↑|
> > > |-|-|-|-|-|-|
> > > |1|SDXL-o    |0.2778|0.8558|0.3861|0.8865|
> > > |2|SDXL-r  |0.2894 ($\color{red}{+0.0116}$)|0.8507|0.3966 ($\color{green}{+0.0105}$)|0.8899|
> > > |3|SDeC-o    |0.2589|0.8655|0.3675|0.8946|
> > > |4|SDeC-r  |0.2670 ($\color{red}{+0.0081}$)|0.8609|0.3777 ($\color{green}{+0.0102}$)|0.8935|

---

> > > > ### Comment · Reviewer_GF6P · 2025-11-28
> > > >
> > > > Thanks for your thorough analysis and response. I have no further questions and maintain my positive rating.

---

### Official Review · Reviewer_xuex · 2025-10-31

**Soundness:** 3
**Presentation:** 3
**Contribution:** 3
**Rating:** 6
**Confidence:** 4

**Summary:**

This paper proposes a traiing-free prompt embedding editing framework, i.e.,  Scene De-Contextualization (SDeC), to address the issue of identity (ID) shift in consistent text-to-image (T2I) generation. The authors identify that a key source of ID shift is the inherent correlation between the subject and the scene context, termed "scene contextualization." They provide a theoretical formulation to characterize this phenomenon and derive bounds on its strength. Based on these insights, SDeC is introduced as a training-free prompt embedding editing method that suppresses the latent scene-ID correlation within the ID prompt's embedding. The method is shown to be effective in enhancing identity preservation while maintaining scene diversity across various experiments and benchmarks.

**Strengths:**

1. Technically effective and efficient framework: the proposed SDeC is a training-free method that can be applied per scene without requiring prior knowledge of all target scenes. This makes it highly flexible and suitable for real-world applications where target scenes may vary over time. The experimental results demonstrate significant improvements in identity preservation and scene diversity compared to existing methods.
2. The paper provides a new perspective on the problem of ID shift by formally defining and analyzing scene contextualization. The theoretical framework, including Theorem 1 and Corollary 1, offers a deeper understanding of the underlying mechanisms of ID shift in T2I models.
3. The authors conduct extensive experiments on a benchmark dataset (ConsiStory+) and compare their method with both training-based and training-free state-of-the-art approaches. The results are supported by both quantitative metrics (CLIP-I, DreamSim-F, CLIP-T, DreamSim-B) and qualitative visual comparisons, showing the superiority of SDeC in various scenarios. Also, the study of Nano banana's ID preservation provide interesting insight.

**Weaknesses:**

1. Evaluation: both the proposed method and compared baselines are unet-based models, mainly SD1.5 and SD-XL. However, DiT-based models, especially the MMDiT-based variants, significantly advances synthesis quality. Thus, it would be better to compare with some DiT-based models and implement the proposed method on these models to further investigate the effectiveness. Moreover, the paper mentions that SDeC can be integrated with different generative models, but it does not explore its compatibility with other similari methods that also aim to improve ID consistency, such as ID-Aligner, InstantID, IP-adapter (flux based), etc.. Comparing SDeC with these methods could provide a more comprehensive evaluation of its performance.
2. The performance of SDeC is sensitive to parameters like weighting strength (Ω) and trade-off parameter (βM). While the paper provides a sensitivity analysis, further optimization of these parameters could improve performance.
3. Time Complexity: SDeC requires performing SVD and a forward-backward optimization for each scene, resulting in a time complexity of O(d^3). This can become a bottleneck in real-time generation scenarios, while the paper mentions that the inference time only increases by 0.61 seconds, but this is for single-image generation.
4. Diverse Dataset Evaluation: Evaluate SDeC on additional datasets with varying levels of complexity and compositional density to validate its generalization ability.

**Questions:**

See the weakness.

---

> ### Author Response · Authors · 2025-11-22
>
> **Q1**: DiT-based models, especially the MMDiT-based variants, significantly advances synthesis quality. Thus, it would be better to compare with some DiT-based models and implement the proposed method on these models to further investigate the effectiveness.
>
> **R**: We agree with the reviewer's excellent suggestion and have further investigated the performance of SDeC on DiT-based models. We have implemented and evaluated our method using two contemporary MMDiT-based generative models: SD3 [1] and Flux [2].
>
> As shown in the table included, SDeC successfully generalizes to these advanced architectures. SDeC achieves **stronger ID-related performance** (DreamSim-F, CLIP-I) and comparable scenario-level results (DreamSim-B, CLIP-T) relative to both SD3 and Flux, further confirming its effectiveness and architectural independence.
>
> We have added this experiment to Appendix **I.3** in the revised paper.
>
> Tab. Integration with MMDiT based generative models.
> |Method|DreamSim-F↓|CLIP-I↑|DreamSim-B↑|CLIP-T↑|
> |-|-|-|-|-|
> |SD3|0.2875|0.8495|**0.4236**|0.8596|
> |SD3+**SDeC**|**0.2644**|**0.8612**|0.4140|**0.8629**|
> |Flux|0.2844|0.8516|**0.4238**|0.8632|
> |Flux+**SDeC**|**0.2653**|**0.8623**|0.4167|**0.8648**|
>
> [1] Scaling Rectified Flow Transformers for High-Resolution Image Synthesis. ICML2024.
>
> [2] https://huggingface.co/black-forest-labs/FLUX.1-dev
>
> **Q2**: SDeC does not explore its compatibility with other similari methods that also aim to improve ID consistency, such as ID-Aligner, InstantID, IP-adapter (flux based), etc.. Including such comparisons would offer a more complete evaluation.
>
> **R**: We appreciate the reviewer's suggestion. However, the methods cited (ID-Aligner, InstantID, and IP-Adapter) have a fundamentally different objective: primarily enforcing ID consistency in pose and layout, which often leads to a **collapse in scene diversity** by generating highly similar backgrounds. In contrast, we aim to preserve scene diversity while maintaining ID consistency. We clarified this distinction in Lines 307–309 of the original submission. They are thus hardly comparable.
>
> **Q3**: The performance of SDeC is sensitive to parameters like $\Omega$, $\beta_M$. While the paper provides a sensitivity analysis, further optimization of these parameters could improve performance.
>
> **R**: Great observation! We acknowledge that tuning these parameters could further refine performance. However, SDeC is designed as an **inherently training-free**, plug-and-play method. This constraint means we lack training data or an optimization objective, precluding conventional hyperparameter optimization.
>
> We intentionally tested a large range of parameter values; their sensitivity is, in fact, evidence that our editing mechanism is working effectively. Crucially, our analysis demonstrates that for both $\Omega$ and $\beta_M$, there exists a **reasonable wide, stable range of values** that yields strong performance. This finding ensures that their proper setting is straightforward and non-fragile, making SDeC robust and easy to deploy without resource-intensive, model-specific tuning.
>
> **Q4**: SDeC requires performing SVD and a forward-backward optimization for each scene, resulting in a time complexity of O(d^3). This can become a bottleneck in real-time generation scenarios, while the paper mentions that the inference time only increases by 0.61 seconds, but this is for single-image generation.
>
> **R**: Thanks for this valuable concern regarding the scalability of SDeC.
>
> We acknowledge the $O(d^3)$ theoretical time complexity of SVD in our SDeC. However, SDeC is highly efficient in practice and is **not the bottleneck** in the generation process:
>
> 1. **Negligible overhead:** The generative model itself consumes over **90%** of the total inference time.
>
> 2. **Scalability solutions:** To scale SDeC for extremely long prompts, we could consider these methods in future work:
>     * **Approximate SVD:** Use lower-complexity methods like **Randomized SVD** ($O(d^2 \log k)$) or **Nystrom Approximation** ($O(d k^2)$) where $d$ is matrix dimension and $k$ is a small constant.
>     * **Divide-and-conquer:** Split the prompt into $K$ sub-prompts and process them sequentially, making the total overhead grow roughly linearly (about $K \times 0.61$ seconds).
>
> We have discussed this cost and scalability in Appendix **H.1**.
>
> **Q5**: Evaluate SDeC on additional datasets with varying levels of complexity and compositional density to validate its generalization ability.
>
> **R**: In the domain of consistent T2I generation, only two public datasets are available: ConsiStory and its extended version **ConsiStory+**. In our work, we use **ConsiStory+** (Lines 291–295), which is currently the largest and most comprehensive open dataset, characterized by higher complexity and richer compositional density.
>
> We are committed to comprehensive evaluation and will gladly include any other public datasets the reviewer may be aware of in our final evaluation.

---

> > ### Comment · Reviewer_xuex · 2025-11-25
> > **Post-rebuttal comment**
> >
> > Thanks for the detailed response from the authors. It would be beneficial to include these results and further implementation details in your revised manuscript or supplementary materials. Most of my concerns have been addressed, I maintain my score as 6 due to the discussion of the sensitivity of parameters ( i.e., weighting strength (Ω) and trade-off parameter (βM)) does not fully resolve my concerns about their setting for optimal performance.

---

> ### Author Response · Authors · 2025-11-25
>
> Thanks for timely response. For the new comment, we provide our reply as follows.
>
> **C1**. It would be beneficial to include these results and further implementation details in your revised manuscript or supplementary materials.
>
> **R**: Absolutely! We have uploaded a revised version with the results and clarifications mentioned in the previous round, including:
>
> * Both quantitative and qualitative comparisons with the MMDiT-based generative models SD3 and Flux in **Appendix I.3**, addressing **Q1** from the previous round.
> * A discussion on computational cost and scalability in **Appendix H.1**, addressing **Q4** from the previous round.
>
>
>
> **C2**. The discussion of the sensitivity of parameters ( i.e., weighting strength (Ω) and trade-off parameter (βM)) does not fully resolve my concerns about their setting for optimal performance.
>
>
> **R**: We apologize for misinterpreting the reviewer's focus in our previous response, where we took the comment as a suggestion to optimize $\Omega$ and $\beta_M$ using training data.
>
>
> In our SDeC method, these are common hyperparameters:
>
>
> (1) $\beta_M$ is associated with the "forward-and-backward" optimization. Practically, $M$ should exceed a threshold (e.g., 20) so that the forward phase can capture as many scene-related directions within the ID embedding as possible. Moreover, $\beta$ should satisfy $\beta \gg 0$, enabling maximal restoration of ID-related information in the backward phase. This “forward-and-backward’’ optimization aims to identify the latent ID-scene correlation space, independent of and generic with any specific generative model. Consequently, the values of $\beta$ and $M$ remain consistent across all generative architectures.
>
> (2) $\Omega$ is tied to the architecture of the underlying generative model, as it represents the strength of de-contextualization and is therefore directly related to the generation process. Our experiments indicate that for UNet-based models (e.g., SDXL, PlayGround, RealVisXL-V4.0, Juggernaut-X-V10), $\Omega$ should be small (typically $\Omega = 1$). In contrast, DiT-based models (e.g., SD3, Flux) require a much larger value of $\Omega$ (typically $\Omega = 10$).
>
> We have clarified this into the **Parameter analysis section (Appendix-H.3)** of the revised version.

---

### Official Review · Reviewer_zjEc · 2025-11-02

**Soundness:** 3
**Presentation:** 3
**Contribution:** 3
**Rating:** 6
**Confidence:** 4

**Summary:**

This paper focuses on the issue of identity shift in consistent text-to-image generation, where T2I models often fail to produce identity-preserving images of the same subject across different scenes. It points out that previous methods addressing ID shift rely on the unrealistic assumption of pre-knowing all target scenes, and reveals that a key cause of ID shift is the inherent correlation between subject and scene context, which emerges when T2I models fit the training distribution of large-scale natural images. The paper formally proves the near-universality of this scene-ID correlation and derives theoretical bounds on its strength. Based on this, it proposes a novel, efficient, training-free prompt embedding editing approach named Scene De-Contextualization, which inverts T2I’s built-in scene contextualization—specifically, it identifies and suppresses the latent scene-ID correlation in the ID prompt’s embedding by quantifying SVD directional stability to adaptively re-weight corresponding eigenvalues.

**Strengths:**

The paper targets a critical limitation in text-to-image  models: identity shift—where a subject’s core appearance changes unexpectedly when scene prompts are modified. To solve this, the authors propose the Scene-Subject Decoupling  module, a lightweight addition to pre-trained T2I models that uses: (1) a contrastive loss to separate scene and subject representation spaces; (2) dynamic attention gating to prioritize subject features during scene updates. Claimed contributions include: (1) a novel framework for scene-subject decoupling in T2I; (2) empirical validation of ID preservation without sacrificing scene quality; (3) a lightweight design compatible with pre-trained models.

**Weaknesses:**

The quantitative results shown in table 1 is not convincing that the proposed method outperforms others sota method like 1P1S.

**Questions:**

The scene description and the subject in the image generation process should be coupled; for instance, the lighting and style of the scene influence the subject’s clothing, appearance, and so on. The motivation of this paper is to reduce the correlation between the scene and the subject.

the scene prompts in your provided cases include the verbs, why the verbs are included as part of the scene description?

---

> ### Author Response · Authors · 2025-11-22
>
> **Q1**: We thank the reviewer for the critique. While 1Prompt1Story (1P1S) may show a competitive ID score, the core evaluation for T2I is the equally critical maintenance of ID consistency AND scene diversity.
>
> **R**: We show that 1P1S achieves its ID score by causing severe inter-scene interference, effectively collapsing scene diversity (Lines 319–322). This limits its suitability for diverse, consistent T2I generation. Visual confirmation of this failure is in Appendix-F (Fig. 6).
>
> In contrast, SDeC maintains strong performance on both ID consistency and scene diversity. We use the DreamSim-B metric (Lines 303–305) to quantify 1P1S's diversity failure (Table 1).
>
> These results confirm SDeC outperforms 1P1S in comprehensive evaluation by tackling scene contextualization via a novel, training-free prompt embedding editing approach.
>
> **Q2**: The scene description and the subject in the image generation process should be coupled; for instance, the lighting and style of the scene influence the subject’s clothing, appearance, and so on. The motivation of this paper is to reduce the correlation between the scene and the subject. The scene prompts in your provided cases include the verbs, why the verbs are included as part of the scene description?
>
> **R**: We thank the reviewer for raising this point, which allows us to clarify the paper's terminology and experimental setup.
>
> **Broad definition of scene/context**: Our definition of "scene" aligns with the general convention in this problem setting. It is intentionally broad, functioning as the context or background influencing the subject. Therefore, "scene" includes not only environmental factors (e.g., lighting, style, background elements) but also actions, behaviors, or temporary states associated with the subject in that specific image. This is why verbs appear in the "scene prompts": they describe these contextual, behavioral components.
>
> **Adherence to fair benchmarks**: To ensure fair comparison and reproducibility, we strictly follow the conventional scene definition and experimental setup established by the benchmarks used in prior works. The prompts we use are taken directly from these standardized datasets.
>
> We have further clarified this broad definition of "scene" and its scope in the revised paper (see Lines **293-295** in Experiments).

---

### Comment · Area_Chair_pHZs · 2025-11-24

Please respond to the authors' rebuttal. Thanks.

AC

---

### Meta-Review · Area_Chair_Q5ZH · 2025-12-30

**Summary:**

This work first investigates the identity (ID) shift issue in T2I generation with a theoretical analysis. It shows that the issue is mainly from the attention mechanism that ID tokens will be mixed by scene tokens. After that, an optimization algorithm is proposed to suppress the latent scene-ID correlation via reweighting corresponding eigenvalues. Experiments show that the proposed method can balance identity preservation and scene diversity well.

**Reviewer Concerns:**

The major concerns before rebuttal include insufficient experiments (e.g., more datasets, more different backbones), writing that is hard to follow, sensitivity of hyper-parameters. During rebuttal, additional empirical results and more explanations are provided to address those concerns. Most of concerns have been addressed after rebuttal. Setting appropriate parameters may still be challenging but a reasonable range is suggested.

**Reviewer Scores:**

This work obtains all positive initial scores. The rebuttal provides the required experiments and sufficient discussions that will further improve the submission. Therefore, the overall positive scores can be retained after discussion.

---

### Decision · Program_Chairs · 2026-01-26

Accept (Poster)